# DyBraSS: Dynamic Brain State Modeling with State-Space Model

## Abstract

Brain states, observable through resting-state functional magnetic resonance imaging (rs-fMRI), represent dynamic transitions between recurring connectivity patterns and are closely linked to neurological and psychiatric conditions. Therefore, developing a computational model for dynamic brain state estimation has been a long-lasting research interest. Among existing approaches, state-space models (SSMs) provide a principled framework for modeling these dynamics. However, existing methods face key limitations: they fail to preserve the brain's spatial architecture, and they model temporal dynamics without considering co-evolving spatial patterns. To address these limitations, we propose **DyBraSS** (**Dy**namic **Bra**in **S**tate-**S**pace model), a novel structured SSM that unifies spatial and temporal modeling within a single framework, enhancing ROI-level modeling capacity and interpretability through a clustering-based global aggregation module. This module respects the brain's network topology by integrating information from all regions during each local update, and represents evolving brain states as interpretable clusters. Comprehensive experiments on multiple fMRI datasets demonstrate that our method consistently outperforms state-of-the-art baselines in diagnostic performance across diverse metrics. Additionally, individual- and group-level brain state analyses reveal that the learned dynamics align with known neurobiological alterations, providing clinically relevant insights for understanding neural dysfunction and developing diagnostic biomarkers.

## 1 Introduction

The human brain exhibits dynamic coordination across distributed networks, with spontaneous fluctuations in functional connectivity (FC) reflecting the flexible organization of neural systems (Sun et al., 2019). Even at rest, the brain transitions through recurring patterns of inter-regional connectivity, termed *brain states*, and resting-state functional magnetic resonance imaging (rs-fMRI) provides noninvasive access to these dynamics via the blood-oxygen-level-dependent (BOLD) signal (Biswal et al., 1995). Extensive research using rs-fMRI has demonstrated that these brain state dynamics are closely linked to neurological and psychiatric conditions (Allen et al., 2014; Lee et al., 2022; Buckner et al., 2008). In particular, neurodevelopmental disorders such as autism spectrum disorder (ASD) (Khan et al., 2013; Jun et al., 2019) and attention-deficit/hyperactivity disorder (ADHD) (Sokunbi et al., 2013) exhibit atypical FCs and transition patterns. These findings underscore the importance of analyzing brain state dynamics for understanding neural dysfunction and developing diagnostic biomarkers.

To investigate brain states, a variety of computational approaches have been proposed, such as clustering-based methods and independent component analysis (ICA) (Van Den Heuvel & Pol, 2010; Beckmann et al., 2005; Calhoun et al., 2001). In particular, state-space models (SSMs) have been widely used, as they provide a principled framework by explicitly describing how latent brain states evolve to generate observed fMRI signals (Friston et al., 2003; Suk et al., 2016). However, conventional SSMs often rely on simplified assumptions such as linearity or fixed parameters, limiting their ability to capture the nonlinear and continuous nature of neural dynamics (Havlicek et al., 2011; Park et al., 2025). Additionally, their computational demands make large-scale and whole-brain analysis challenging in practice (Zhang et al., 2019; Prando et al., 2020). These limitations have motivated the development of more flexible modeling approaches that effectively capture complex nonlinear dynamics while maintaining the theoretical advantages of state-space formulations.

Recently, advances in structured sequence modeling have given rise to new possibilities for capturing complex and continuous patterns while maintaining computational scalability (Bansal et al., 2024; Heidari et al., 2024). In particular, by leveraging efficient parameterizations and selective mechanisms from models such as S4 (Gu et al., 2022c) and Mamba (Gu & Dao, 2024), neuroimaging applications have achieved state-of-the-art (SOTA) performance in clinical prediction tasks (El-Gazzar et al., 2022; Behrouz & Hashemi, 2024; Wei et al., 2025; Wang et al., 2025). Despite this progress, fundamental challenges remain in applying these structured SSMs to brain dynamics. Due to the inherently temporal formulation of SSMs, current methods often impose arbitrary sequential orderings on brain regions (*i.e.*, regions of interest [ROIs]), which disregards the brain's intrinsic topological organization. Furthermore, spatial and temporal dynamics are typically modeled in isolation and only integrated at later stages, constraining the ability to capture complex, co-evolving connectivity patterns across distributed brain networks.

To address these limitations, we propose the **Dy**namic **Bra**in **S**tate-**S**pace model (**DyBraSS**), which is based on a novel structured SSM formulation that integrates spatial and temporal modeling of brain dynamics within a unified framework, enhancing ROI-level modeling capacity and interpretability. Central to our design is a global aggregation module that preserves the brain's network topology by incorporating information from all brain regions into each local (*i.e.*, ROI-level) temporal update. To further capture higher-order spatial dependencies, *i.e.*, *co-evolving spatial patterns*, we employ a clustering-based strategy that represents dynamic brain states as interpretable clusters, each corresponding to a distinct connectivity configuration. This architecture enables the modeling of brain state transitions while facilitating clinical interpretation by revealing evolving patterns across diagnostic groups. To evaluate the effectiveness and generalizability of the proposed framework, we conducted comprehensive experiments on multiple fMRI datasets. Compared with SOTA baselines, our method achieved superior performance across diverse metrics. Furthermore, in-depth analyses, including individual- and group-level brain state analyses and ablation studies, demonstrated the expressive power of our model and validated its robustness.

In summary, our main contributions are as follows:

- We propose DyBraSS, a novel SSM-based model that jointly models inter-ROI interactions during temporal state evolution. By using a global aggregation mechanism, our approach ensures that each ROI's dynamics incorporate contextual information from all brain regions, preserving the brain's topological structure.

- We integrate a clustering-based approach within the aggregation module to represent latent brain states as interpretable clusters. This enables explicit modeling of evolving brain state trajectories and enhances clinical interpretability.

- Our model demonstrates effectiveness in diagnosing ASD and ADHD from rs-fMRI data, outperforming SOTA baselines. Furthermore, the learned dynamic brain states align with known neurobiological alterations, providing potential insights for neuroscience and clinical applications.

## 2 RELATED WORK

### 2.1 BRAIN NETWORK ANALYSIS

Deep learning approaches for brain network analysis have adopted diverse architectures to capture the complex spatiotemporal patterns of neural systems (Wen et al., 2018; Yin et al., 2022; Jeong et al., 2024). For example, convolutional neural network (CNN)-based methods such as BrainNetCNN (Kawahara et al., 2017) leverage specialized convolutional filters for brain connectomes, exploiting topological locality in structural connectivity. Graph-based approaches like BrainGNN (Li et al., 2021) introduce a graph neural network (GNN) framework with ROI-aware convolutions on fMRI-derived networks to learn interpretable connectivity biomarkers. More recently, building on the success of attention mechanisms (Vaswani et al., 2017), Transformer-based approaches have further advanced the field. BrainNetTF (Kan et al., 2022) employs a graph Transformer with an orthonormal clustering-based readout, pooling brain regions into functional modules to produce discriminative graph embeddings. BolT (Bedel et al., 2023) uses a fused-window attention mechanism that hierarchically grows overlapping time windows to capture both local and global BOLD signal patterns. While these methods have demonstrated notable success, they primarily model static

brain networks or use discrete temporal representations, limiting their ability to capture continuous temporal dynamics of neural systems.

## 2.2 State-Space Models for Brain Dynamics

Conventional SSMs have been employed to describe neural processes underlying neuroimaging data due to their ability to capture temporal evolution of brain activity through latent state dynamics. Dynamic causal modeling (DCM) estimates effective connectivity between brain regions through Bayesian inference on continuous state-space formulations (Friston et al., 2003). Kalman filter-based models (Wang et al., 2023; Havlicek et al., 2011) treat brain activity as a continuously evolving latent state, using recursive filtering to capture varying connectivity changes over time. Hidden Markov models (HMMs) (Suk et al., 2016; Zhang et al., 2019) assume the brain transitions among a discrete set of latent states, each characterized by a distinct pattern of activity or connectivity. However, these conventional approaches often face computational scalability issues or rely on simplified linear or discrete assumptions that may limit their ability to capture complex brain dynamics.

Recently, structured SSMs such as S4 (Gu et al., 2022c) and Mamba (Gu & Dao, 2024) have emerged, enabling efficient learning of long-range dependencies with linear complexity. Building on this, several variants have been proposed for neuroimaging applications: fMRI-S4 (El-Gazzar et al., 2022) integrates convolutional layers with S4 modules to jointly model short- and long-range temporal dependencies in rs-fMRI. Brain-Mamba (Behrouz & Hashemi, 2024) integrates selective S4 encoders to model temporal dynamics with GNNs for brain region selection. FST-Mamba (Wei et al., 2025) adopts a hierarchical design tailored to dynamic FC (dFC) matrices, incorporating component-wise aggregation and positional encoding. BrainMAP (Wang et al., 2025) employs a mixture-of-experts framework to sequentialize brain graphs and extract multiple activation pathways. Together, these models illustrate how structured sequence modeling can provide scalable and flexible representations of brain state dynamics. However, existing structured SSM approaches for fMRI analysis face important limitations. They typically either treat ROIs as sequential inputs with arbitrary ordering or do not directly model relationships between regions, failing to capture the brain's spatial topology and global interactions during temporal evolution. To address these limitations, we propose a novel structured SSM formulation that explicitly models inter-ROI relationships while leveraging selective mechanisms (Gu & Dao, 2024) in a parameter-efficient manner.

## 3 Preliminaries

**Structured state-space models.** Continuous-time linear ordinary differential equations (ODEs) have been widely used to model sequential data, forming the basis of structured SSMs that enable efficient long-range sequence processing (Gu et al., 2021; 2022b). Consider a 1-dimensional input sequence $x(t) \in \mathbb{R}$ and output $y(t) \in \mathbb{R}$. The hidden state $\mathbf{h}(t) \in \mathbb{R}^N$, where $N$ denotes the state-space dimension, evolves under linear transitions, and the output is obtained as follows:

$$\frac{d}{dt}\mathbf{h}(t) = \mathbf{A}\mathbf{h}(t) + \mathbf{B}x(t), \tag{1}$$

$$y(t) = \mathbf{C}\mathbf{h}(t) + \mathbf{D}x(t), \tag{2}$$

where $\mathbf{A} \in \mathbb{R}^{N \times N}$ is the state-transition matrix, $\mathbf{B} \in \mathbb{R}^{N \times 1}$, $\mathbf{C} \in \mathbb{R}^{1 \times N}$ are projection matrices, and $\mathbf{D} \in \mathbb{R}$ is a residual connection parameter. For multi-dimensional inputs, this formulation can be extended by applying independent SSMs to each feature dimension, with separate transition matrices and hidden states for each dimension (*e.g.*, $\mathbf{A} \in \mathbb{R}^{D \times N \times N}$, $\mathbf{h} \in \mathbb{R}^{D \times N}$ for $D$-dimensional input).

Recent models such as Mamba (Gu & Dao, 2024) improve adaptability to varying inputs by making parameters $\mathbf{A}, \mathbf{B}, \mathbf{C}$ and the discretization step size depending on the input. Specifically, the system is discretized using zero-order hold (ZOH) and takes the form at time step $t$:

$$\mathbf{h}_t = \bar{\mathbf{A}}_t\mathbf{h}_{t-1} + \bar{\mathbf{B}}_t x_t, \tag{3}$$

$$y_t = \mathbf{C}_t\mathbf{h}_t + \mathbf{D}x_t, \tag{4}$$

where $\bar{\mathbf{A}}_t = \exp(\Delta_t \mathbf{A})$ and $\bar{\mathbf{B}}_t = \mathbf{A}^{-1}(\bar{\mathbf{A}}_t - \mathbf{I})\mathbf{B}_t$, with $\Delta_t$ denoting the input-dependent sampling interval. This formulation allows the model to selectively control information flow in a data-dependent manner. In the next section, building on this selective structured SSM, we propose a method specifically designed for dynamic brain state modeling.

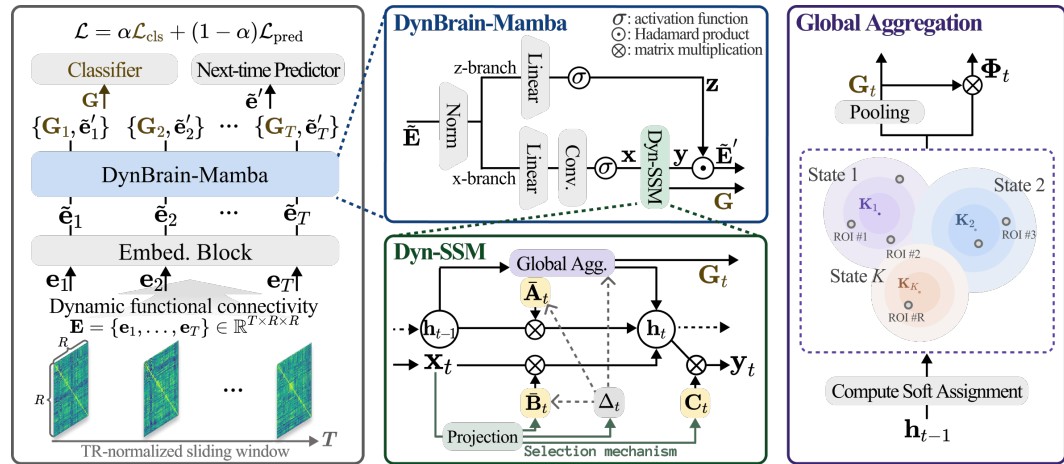

Figure 1: The overall workflow of the proposed DyBraSS. The input sequence of FC matrices (*i.e.*, dFC) is first embedded and then processed by the DynBrain-Mamba block, which produces (i) a global embedding $\mathbf{G}$ for clinical diagnosis and (ii) refined ROI representations $\tilde{\mathbf{e}}'$ for next-time FC prediction. This block, based on the Dyn-SSM, jointly models the temporal evolution of ROIs while incorporating global brain context via a clustering-based global aggregation module that organizes ROI-level features into brain state clusters. Through this architecture, DyBraSS enables spatiotemporal modeling and captures evolving brain state transitions, enhancing clinical interpretability.

## 4 METHODOLOGY

The overall structure of our proposed model is illustrated in Fig. 1. Our approach takes a sequence of FC matrices (*i.e.*, dFC) as input, and an embedding block maps each FC matrix to ROI-wise representations. These embeddings are processed by a dynamic brain Mamba (DynBrain-Mamba) block based on dynamic-SSM (Dyn-SSM), which models the temporal dynamics of each ROI while incorporating inter-ROI context through a global aggregation module. Within this module, a cluster-based mechanism organizes ROI representations into brain state clusters, and this global context is integrated into local ROI processing, enabling spatiotemporal modeling of evolving brain dynamics. Finally, the resulting outputs from DynBrain-Mamba, including global features and ROI-wise sequences, are processed by task-specific heads for clinical diagnosis and next-time FC prediction.

### 4.1 EMBEDDING FOR DYNAMIC FUNCTIONAL CONNECTIVITY

To construct the input representation, we compute dFC from BOLD signals using a sliding-window approach (Allen et al., 2014). Specifically, within each temporal window, we calculate pairwise Pearson correlations between ROI signals to obtain an FC matrix. To ensure temporal consistency across different acquisition sites with varying repetition times, the window length and stride are normalized as $w = \lfloor w_{\text{sec}}/\text{TR} \rfloor$ and $s = \lfloor s_{\text{sec}}/\text{TR} \rfloor$, respectively, where $w_{\text{sec}}$ and $s_{\text{sec}}$ are the target window length and stride in seconds (TR; the time interval between successive volume acquisitions). This formulation ensures that each time step corresponds to a consistent temporal unit regardless of TR variability across datasets. The resulting sequence of FC matrices is denoted as $\mathbf{E} = \{\mathbf{e}_t\}_{t=1}^T \in \mathbb{R}^{T \times R \times R}$, where $\mathbf{e}_t$ represents the FC matrix at time window $t$ of the total $T$ windows, and $R$ is the number of ROIs.

Each FC matrix $\mathbf{e}_t \in \mathbb{R}^{R \times R}$ is then transformed to ROI-wise embeddings $\tilde{\mathbf{e}}_t$ by mapping each ROI's connectivity profile through an embedding block implemented as a multi-layer perceptron (MLP):

$$\widetilde{\mathbf{E}} = \{\tilde{\mathbf{e}}_t\}_{t=1}^T = \text{Embed}(\mathbf{E}) \in \mathbb{R}^{T \times R \times D}, \qquad (5)$$

where $D$ is the embedding dimension, and the resulting embedded sequences serve as input to the subsequent DynBrain-Mamba block.

### 4.2 DynBrain-Mamba: Spatiotemporal Modeling of Dynamic Brain States

Brain dynamics are characterized by coordinated interactions among distributed regions, where ROIs continuously influence each other's activity patterns rather than evolving in isolation (Sun et al., 2019). Based on this principle, we propose a novel DynBrain-Mamba that explicitly captures brain state trajectories by modeling inter-ROI spatial dependencies during temporal transitions.

Specifically, as illustrated in Fig. 1, given the embedded sequence $\widetilde{\mathbf{E}}$, this block first applies normalization and linear projections, forming two parallel processing streams: an x-branch for main representation learning and a z-branch for gating. The x-branch is passed through a temporal convolution to produce intermediate features $\mathbf{x} \in \mathbb{R}^{T \times R \times D}$, which are then processed by the Dyn-SSM. This module models temporal dynamics while explicitly capturing brain state evolution through clustering-based global aggregation, yielding (1) global embeddings $\mathbf{G}$ and (2) ROI-wise features $\mathbf{y}$. The ROI-wise features are gated via a Hadamard product with the z-branch output $\mathbf{z} \in \mathbb{R}^{T \times R \times D}$, producing the final output sequence $\widetilde{\mathbf{E}}'$. The details of the proposed Dyn-SSM are described below, and its global aggregation method is explained in Section 4.2.2.

#### 4.2.1 Dynamic Brain State Modeling with Global Context

**Dyn-SSM formulation.** The Dyn-SSM extends standard structured SSMs (Eqs. 1–2) to each ROI, enhancing ROI-specific modeling capacity while incorporating a global aggregation mechanism that allows access to contextual information from all other brain regions during state evolution.

Given a sequence $\{\mathbf{x}_r(t)\}_{r=1}^R$, where $\mathbf{x}_r(t) \in \mathbb{R}^D$ represents the input of the $r$-th ROI obtained from the x-branch, we define a hidden state $\mathbf{h}_r(t) \in \mathbb{R}^N$, where $N$ denotes the state-space dimension, for each ROI. The evolution of $\mathbf{h}_r(t)$ is governed by both its local input $\mathbf{x}_r(t)$ and the globally aggregated information across all ROIs as follows:

$$\frac{d}{dt}\mathbf{h}_r(t) = \mathbf{A}_r \mathbf{h}_r(t) + \mathbf{B}_r \mathbf{x}_r(t) + \boldsymbol{\Phi}_r(t), \tag{6}$$

$$\mathbf{y}_r(t) = \mathbf{C}_r \mathbf{h}_r(t) + \mathbf{D}_r \odot \mathbf{x}_r(t), \tag{7}$$

where $\mathbf{A}_r \in \mathbb{R}^{N \times N}$ is the transition matrix, $\mathbf{B}_r \in \mathbb{R}^{N \times D}$ and $\mathbf{C}_r \in \mathbb{R}^{D \times N}$ are projection matrices, $\mathbf{D}_r \in \mathbb{R}^D$ is the skip connection parameter, and $\odot$ denotes the Hadamard product. The term $\boldsymbol{\Phi}_r(t)$ denotes the $r$-th component of $\boldsymbol{\Phi}(t) := \mathcal{G}\left(\{\mathbf{h}_i(t)\}_{i=1}^R\right)$, where $\mathcal{G}(\cdot)$ is the global aggregation function that incorporates spatial context by organizing ROI hidden states into brain state representations through a clustering-based mechanism (detailed in Section 4.2.2).

Note that we define our SSM formulation with unified hidden states $\mathbf{h}_r(t) \in \mathbb{R}^N$ and transition matrices $\mathbf{A}_r \in \mathbb{R}^{N \times N}$ for the $r$-th ROI, using projections $\mathbf{B}_r$ and $\mathbf{C}_r$ to integrate the $D$-dimensional input $\mathbf{x}_r$ for each ROI, rather than maintaining separate parameters for each feature dimension as in standard formulations mentioned in Section 3.

**Discretization.** The continuous-time formulation is discretized using ZOH and the variation of constants formula. The resulting hidden-state update for the $r$-th ROI at time step $t$ takes the form:

$$\mathbf{h}_{r,t} = \bar{\mathbf{A}}_r \mathbf{h}_{r,t-1} + \bar{\mathbf{B}}_r \mathbf{x}_{r,t} + \bar{\boldsymbol{\Phi}}_{r,t}, \tag{8}$$

$$\mathbf{y}_{r,t} = \mathbf{C}_r \mathbf{h}_{r,t} + \mathbf{D}_r \odot \mathbf{x}_{r,t}, \tag{9}$$

where the discretized parameters are defined using the step size $\Delta_r$ as follows:

$$\bar{\mathbf{A}}_r = \exp\left(\Delta_r \mathbf{A}_r\right), \quad \bar{\mathbf{B}}_r = \mathbf{A}_r^{-1}\left(\bar{\mathbf{A}}_r - \mathbf{I}\right) \mathbf{B}_r, \quad \bar{\boldsymbol{\Phi}}_{r,t} = \mathbf{A}_r^{-1}\left(\bar{\mathbf{A}}_r - \mathbf{I}\right) \boldsymbol{\Phi}_{r,t}. \tag{10}$$

Here, $\boldsymbol{\Phi}_{r,t}$ denotes the $r$-th row of $\boldsymbol{\Phi}_t$. Detailed derivations are provided in Appendix A.

**Input-dependent parameterization.** Following the selective mechanism in Mamba (Gu & Dao, 2024), we make the parameters input-dependent by defining them as:

$$\mathbf{B}_{r,t} = f_B(\mathbf{x}_{r,t}), \quad \mathbf{C}_{r,t} = f_C(\mathbf{x}_{r,t}), \quad \Delta_{r,t} = \text{softplus}\left(\epsilon_r + f_\Delta(\mathbf{x}_{r,t})\right), \tag{11}$$

where $f_B, f_C$, and $f_\Delta$ are linear projections, and $\epsilon_r \in \mathbb{R}$ is a learnable parameter. This design dynamically generates projection matrices from the input, reducing computational cost while preserving the spatial topology and individual ROI modeling capacity. In addition, the input-conditioned $\Delta_{r,t}$ together with the ROI-specific parameter $\epsilon_r$ provides selective control of the dynamics at the ROI level, rather than embedding dimension-wise selectivity in standard formulations, enabling the model to adaptively focus on both relevant brain regions and time points.

### 4.2.2 ORTHONORMAL CLUSTER AGGREGATION

Clustering-based approaches have played a central role in fMRI studies, as they provide a principled way to identify recurring brain states by grouping time-varying connectivity patterns into discrete clusters that reflect the brain's inherent functional modules (Allen et al., 2014; Damaraju et al., 2014; Liu et al., 2012). Motivated by this insight, we incorporate a clustering-based method into our SSM formulation, aggregating ROI hidden states at each time step. This design enables the model to track evolving brain states while simultaneously obtaining global-level embeddings that are fed back to update each ROI's hidden state, thereby capturing both temporal dynamics and spatial dependencies across ROIs. Details of the aggregation mechanism are described below.

**Orthonormal clustering.** Prior studies have shown that orthonormal bases can effectively decompose brain representations into clustered connectivity patterns corresponding to interpretable network modes (Kan et al., 2022; Caffo et al., 2010). Building on this property, we adopt an orthonormal cluster aggregation to track brain dynamics while integrating global information, where each cluster represents a distinct brain state, and ROIs are softly assigned to clusters at each time step.

Specifically, we define $K$ cluster centers as orthonormal bases in $\mathbb{R}^N$, denoted as $\mathbf{K} \in \mathbb{R}^{K \times N}$. These bases are derived via the Gram-Schmidt process:

$$\mathbf{u}_k = \mathbf{V}_k - \sum_{j=1}^{k-1} \frac{\langle \mathbf{u}_j, \mathbf{V}_k \rangle}{\langle \mathbf{u}_j, \mathbf{u}_j \rangle} \mathbf{u}_j, \quad \mathbf{K}_k = \frac{\mathbf{u}_k}{||\mathbf{u}_k||}, \tag{12}$$

where $\langle \cdot, \cdot \rangle$ denotes the inner product, and $\mathbf{V} \in \mathbb{R}^{K \times N}$ are the initial random vectors generated using Xavier uniform initialization. Given these orthonormal cluster centers, we compute similarity scores with the hidden state from the previous time step and define the assignment probability that ROI $r$ is assigned to cluster $k$ at time $t$ as:

$$\pi_{r,k,t} = \frac{\exp(\langle \mathbf{h}_{r,t-1}, \mathbf{K}_k \rangle)}{\sum_{k'=1}^{K} \exp(\langle \mathbf{h}_{r,t-1}, \mathbf{K}_{k'} \rangle)}, \quad \mathbf{P}_t = \begin{bmatrix} \pi_{1,1} & \cdots & \pi_{1,K} \\ \vdots & \ddots & \vdots \\ \pi_{R,1} & \cdots & \pi_{R,K} \end{bmatrix}_t. \tag{13}$$

These probabilities represent the soft assignment of each ROI to brain state clusters, where ROIs with similar hidden representations receive higher assignment probabilities to the same clusters.

Using the assignment matrix $\mathbf{P}_t$, we aggregate ROI-level representations $\mathbf{h}_{t-1} = [\mathbf{h}_{r,t-1}]_{r=1}^{R} \in \mathbb{R}^{R \times N}$ into global-level embeddings through assignment-weighted sum pooling:

$$\mathbf{G}_t = \mathbf{P}_t^{\top} \mathbf{h}_{t-1} \in \mathbb{R}^{K \times N}. \tag{14}$$

The global embeddings $\mathbf{G}_t$ capture brain state information aggregated from all ROIs and are redistributed to ROI-level representations $\mathbf{\Phi}_t$ based on the same soft assignment probabilities:

$$\mathbf{\Phi}_t = \mathbf{P}_t \mathbf{G}_t \in \mathbb{R}^{R \times N}. \tag{15}$$

This term is incorporated into the Dyn-SSM equations (Eq. 8) after discretization, enabling the model to incorporate global context while modeling brain state transitions over time.

### 4.3 DECODING HEADS AND OBJECTIVES

The outputs of the DynBrain-Mamba block, the global embeddings $\{\mathbf{G}_t\}_{t=1}^{T}$ and the output sequence $\widetilde{\mathbf{E}}'$, are used for distinct tasks. The global embeddings $\{\mathbf{G}_t\}_{t=1}^{T} \in \mathbb{R}^{T \times K \times N}$ are processed by a convolution block consisting of two 2D convolutional layers along the time axis to capture evolving patterns for diagnostic classification, optimized with cross-entropy loss $\mathcal{L}_{\text{cls}}$. Concurrently, the output sequence $\widetilde{\mathbf{E}}' \in \mathbb{R}^{T \times R \times D}$ is passed to an MLP, where each time step feature $\tilde{\mathbf{e}}'_t$ is used to predict the next-time FC with mean absolute error loss $\mathcal{L}_{\text{pred}}$.

The final training objective function is defined with a weighting hyperparameter $\alpha$ as follows:

$$\mathcal{L} = \alpha \mathcal{L}_{\text{cls}} + (1 - \alpha) \mathcal{L}_{\text{pred}}. \tag{16}$$

This multi-task learning framework enables the model to capture subject-level dynamics while simultaneously optimizing group-level diagnostic prediction. During inference, only the global embeddings $\{\mathbf{G}_t\}_{t=1}^{T}$ are used for diagnostic classification.

Table 1: Comparison with baseline models for classification on ABIDE-I and ADHD-200 datasets.

| Method | ABIDE-I | | | | ADHD-200 | | | |
|---|---|---|---|---|---|---|---|---|
| | AUROC | ACC (%) | SEN (%) | SPC (%) | AUROC | ACC (%) | SEN (%) | SPC (%) |
| BrainNetCNN | $0.6441\pm0.02^*$ | $64.32\pm1.71^*$ | $48.61\pm7.58^*$ | $75.93\pm3.93$ | $0.6379\pm0.03$ | $62.67\pm2.75^*$ | $52.02\pm8.65^*$ | $62.66\pm8.22$ |
| BrainNetTF | $0.6544\pm0.02^*$ | $64.02\pm0.75^*$ | $54.55\pm11.89^*$ | $71.54\pm9.06$ | $0.6325\pm0.01^*$ | $63.14\pm2.76^*$ | $52.55\pm5.64$ | $66.20\pm1.69$ |
| BoIT | $0.6440\pm0.04^*$ | $63.44\pm2.39^*$ | $61.72\pm5.49$ | $58.85\pm12.43$ | $0.6294\pm0.02^*$ | $62.52\pm2.18^*$ | $53.19\pm7.08$ | $64.98\pm9.00$ |
| ContrastPool | $0.6332\pm0.02^*$ | $62.30\pm2.19^*$ | $54.40\pm4.30$ | $64.41\pm6.24$ | $0.6151\pm0.01^*$ | $59.60\pm2.10^*$ | $46.65\pm10.76^*$ | $60.39\pm8.51$ |
| fMRI-S4 | $0.6483\pm0.02^*$ | $63.59\pm2.70^*$ | $40.94\pm7.06^*$ | $\mathbf{79.47\pm8.81}$ | $0.6470\pm0.02^*$ | $63.30\pm1.91^*$ | $53.42\pm10.64^*$ | $\mathbf{70.03\pm8.54}$ |
| FST-Mamba | $0.6284\pm0.04^*$ | $62.51\pm3.35^*$ | $43.54\pm9.84^*$ | $74.46\pm10.19$ | $0.6157\pm0.01^*$ | $61.73\pm2.86^*$ | $48.47\pm6.73^*$ | $69.63\pm5.98$ |
| BrainMAP | $0.6625\pm0.03$ | $64.47\pm3.18^*$ | $49.20\pm8.10^*$ | $75.34\pm7.96$ | $0.6508\pm0.03^*$ | $64.47\pm2.57$ | $51.84\pm7.61^*$ | $68.61\pm10.68$ |
| DyBraSS (Ours) | $\mathbf{0.6904\pm0.01}$ | $\mathbf{67.99\pm2.53}$ | $\mathbf{63.38\pm6.98}$ | $68.27\pm7.13$ | $\mathbf{0.6727\pm0.01}$ | $\mathbf{66.23\pm2.34}$ | $\mathbf{60.93\pm9.41}$ | $62.24\pm5.91$ |

Best scores are highlighted in **bold**, second best are underline, and * indicates statistical significance ($p < 0.05$).

## 5 EXPERIMENTS

**Datasets.** We conducted experiments on two publicly available rs-fMRI datasets: Autism Brain Imaging Data Exchange (ABIDE)-I (Di Martino et al., 2014) and ADHD-200 (consortium, 2012). Following preprocessing, we retained 678 subjects from ABIDE-I (290 ASD; 388 typical controls, TC) and 643 from ADHD-200 (270 ADHD; 373 typically developing, TD). The brain was parcellated into 114 ROIs using the Yeo 2011 atlas (Yeo et al., 2011). Details are provided in Appendix D.

**Baselines.** We compared against fMRI-based SOTA methods spanning multiple architectural paradigms: BrainNetCNN (Kawahara et al., 2017) for CNN-based methods, BrainNetTF (Kan et al., 2022) and BoIT (Bedel et al., 2023) for Transformer-based methods, ContrastPool (Xu et al., 2024) for graph-based methods, and fMRI-S4 (El-Gazzar et al., 2022), FST-Mamba (Wei et al., 2025), and BrainMAP (Wang et al., 2025) for structured SSM-based methods. We used official implementations with hyperparameters tuned for each method. Further details are provided in Appendix E.2.

**Experimental setup.** We performed 5-fold cross-validation with splits of 70% training, 10% validation, and 20% testing, with identical splits across baselines. Performance was evaluated with area under the receiver operating characteristic curve (AUROC), accuracy (ACC), sensitivity (SEN), and specificity (SPC). The best model was selected by the highest validation AUROC, and results were averaged over all test folds; statistical significance ($*: p < 0.05$) was verified via paired $t$-tests. Implementation details are provided in Appendix E.1 and our GitHub[1].

### 5.1 MAIN RESULTS

The comparison results with baseline methods are presented in Table 1. SSM-based methods, specifically DyBraSS and BrainMAP, achieve the highest and second-highest AUROC and ACC on both datasets. Particularly on ADHD-200, the top-performing methods (DyBraSS, BrainMAP, and fMRI-S4) are all SSM-based, highlighting the effectiveness of structured sequence modeling for clinical diagnosis. Among them, our model consistently attains the best AUROC, ACC, and SEN on both datasets, improving AUROC and ACC by approximately 2–4 percentage points over the next-best baseline (*i.e.*, BrainMAP) while maintaining balanced SEN and SPC. Overall, these results show that DyBraSS achieves SOTA performance across multiple datasets and evaluation metrics, demonstrating the effectiveness of our method. Additional comparative results on the COBRE dataset (Mayer et al., 2013), demonstrating the robustness of our framework, are provided in Appendix F.3.2.

### 5.2 ABLATION STUDIES

**Influence of aggregation mechanisms.** To validate our aggregation strategy (Orth.), we conducted ablation studies comparing several alternatives: removing the aggregation module (w/o Agg), using other aggregation methods (Mean, Sum, Attention), and clustering-based aggregation with different initialization schemes (Random: fixed randomly initialized centers, Random‡: learnable random centers, initialized with Xavier uniform distribution). As shown in Table 2, removing the aggregation module resulted in performance degradation in AUROC and ACC, confirming the necessity of inter-ROI context integration. Other aggregation methods showed modest gains over w/o Agg but

---

[1] https://anonymous.4open.science/r/DyBraSS-8714

Table 2: Ablation study of global aggregation mechanisms.

| Method | ABIDE-I | | | | ADHD-200 | | | |
|---|---|---|---|---|---|---|---|---|
| | AUROC | ACC (%) | SEN (%) | SPC (%) | AUROC | ACC (%) | SEN (%) | SPC (%) |
| w/o Agg | 0.6097±0.05* | 61.52±2.54* | 42.23±6.29* | **75.29±7.62** | 0.6169±0.01* | 60.45±2.73* | 49.43±8.47* | 71.45±8.62 |
| Mean | 0.6256±0.04* | 62.37±4.24* | 49.67±6.79* | 68.36±8.62 | 0.6220±0.04* | 62.81±5.56* | 43.06±6.47* | 74.40±9.77* |
| Sum | 0.6114±4.39* | 61.88±2.81* | 42.57±5.20* | 72.72±4.56 | 0.6207±0.02* | 62.38±2.85* | 44.26±6.32* | 71.81±5.31* |
| Attention | 0.6331±0.03* | 63.06±3.70* | 42.38±4.54* | 73.78±8.87 | 0.6216±0.02* | 62.51±2.43* | 33.84±9.52* | **74.87±14.25** |
| Random | 0.6583±0.02* | 64.75±3.23* | 54.90±6.44 | 72.18±6.53 | 0.6361±0.02* | 63.36±1.96* | 49.63±5.03* | 67.57±7.86 |
| Random‡ | 0.6587±0.03* | 65.02±2.92* | 54.69±9.34* | 67.55±17.18 | 0.6391±0.02* | 63.41±1.34* | 50.66±5.84 | 68.75±7.85 |
| Ours | **0.6904±0.01** | **67.99±2.53** | **63.38±6.98** | 68.27±7.13 | **0.6727±0.01** | **66.23±2.34** | **60.93±9.41** | 62.24±5.91 |

Best scores are highlighted in **bold**, second best are underline, and * indicates statistical significance ($p < 0.05$).

Table 3: Ablation on next-time FC prediction loss ($\mathcal{L}_{\mathrm{pred}}$) and site-specific TR normalization.

| Method | ABIDE-I | | | | ADHD-200 | | | |
|---|---|---|---|---|---|---|---|---|
| | AUROC | ACC (%) | SEN (%) | SPC (%) | AUROC | ACC (%) | SEN (%) | SPC (%) |
| w/o $\mathcal{L}_{\mathrm{pre}}$ | 0.6514±0.02* | 64.32±2.95* | 48.78±7.30* | **74.64±4.88** | 0.6451±0.02* | 64.54±1.83* | 47.04±6.74* | **74.46±4.49*** |
| w/o TR | 0.6640±0.03 | 65.10±2.24* | 61.11±8.75* | 66.19±11.74 | 0.6550±0.01* | 65.16±2.10* | 52.18±8.64 | 70.16±8.98* |
| Ours | **0.6904±0.01** | **67.99±2.53** | **63.38±6.98** | 68.27±7.13 | **0.6727±0.01** | **66.23±2.34** | **60.93±9.41** | 62.24±5.91 |

Best scores are highlighted in **bold**, second best are underline, and * indicates statistical significance ($p < 0.05$).

underperformed relative to clustering methods across AUROC, ACC, and SEN. All clustering-based approaches consistently outperformed other methods, while our orthonormal clustering attained the highest AUROC, ACC, and SEN, underscoring its effectiveness for clinical diagnosis.

**Influence of next-time FC prediction task.** To evaluate the impact of next-time FC prediction task on diagnostic classification performance, we removed the prediction MLP in decoding head and excluded $\mathcal{L}_{\mathrm{pred}}$ from training. As shown in Table 3, excluding $\mathcal{L}_{\mathrm{pred}}$ (w/o $\mathcal{L}_{\mathrm{pred}}$) resulted in reduced AUROC, ACC, and SEN on both datasets, indicating that the auxiliary prediction task contributes to more robust diagnostic performance.

**Influence of site-specific TR normalization.** We assessed the effect of handling TR variations across acquisition sites by comparing our approach with a baseline that sets a uniform TR of 1s across all sites (w/o TR). As shown in Table 3, disabling TR normalization yielded lower AUROC and ACC on both datasets. The degradation was more pronounced on ABIDE-I, which may be attributed to its larger number of acquisition sites (17 sites) compared to ADHD-200 (8 sites), resulting in greater site variability. This underscores the importance of properly handling site-related variations when modeling brain dynamics in multi-site neuroimaging studies.

## 5.3 BRAIN STATE ANALYSIS

We analyzed brain state dynamics using importance scores computed with Captum (Kokhlikyan et al., 2020), conducting both individual- and group-level analyses to examine temporal evolution and group differences. Further implementation details are provided in Appendix F.1.

**Individual-level analysis.** To examine temporal brain state dynamics at the individual-level, we visualized the top-1 cluster by importance at each time step for representative subjects from each group in the ABIDE-I dataset (Fig. 2). The visualizations reveal that brain state transitions with distinct patterns across subjects, highlighting our model's capacity to capture subject-specific temporal dynamics. Moreover, the trends of their importance scores align with group-discriminative transition patterns observed in the group-level analysis below. This demonstrates that our integrated framework, which combines a structured SSM with clustering-based aggregation,

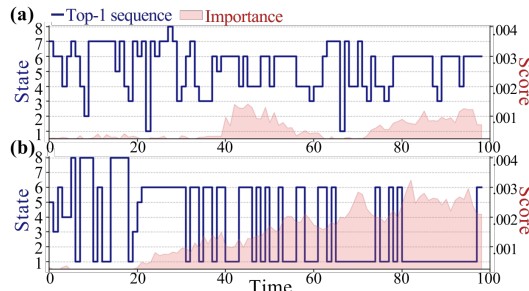

Figure 2: Brain state evolution with top-1 cluster sequence (blue) and importance scores (pink) for (a) an ASD and (b) a TC case from ABIDE-I.

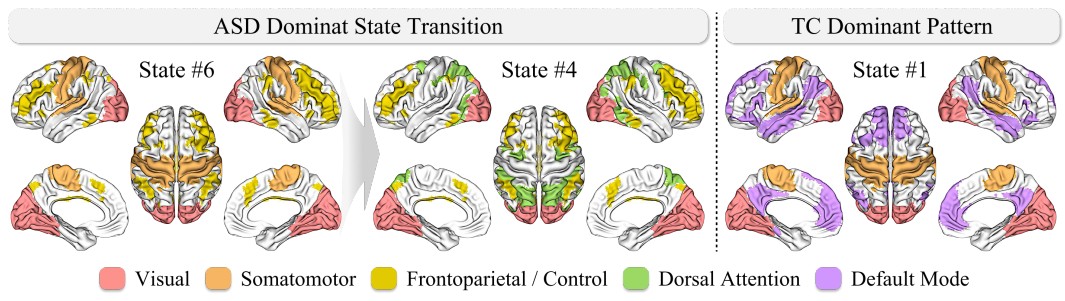

Figure 4: Brain network configurations of discriminative states in ABIDE-I.

not only tracks individual-level trajectories but also learns patterns that are discriminative between groups. Additional representative samples and ADHD-200 cases are provided in Appendix F.2.

**Group-level analysis.** To analyze group differences, we constructed brain state transition maps for each diagnostic group and examined the brain network assignments for each state, validating the observed transition patterns using independent $t$-tests (results in Appendix F.3). Fig. 3 shows the transition matrix differences in ABIDE-I, highlighting significant patterns ($p < 0.001$) where ASD exhibited stronger State 6→4 transitions and TC showed a greater tendency to remain in State 1.

To interpret these discriminative patterns, we visualized the network compositions of key states in Fig. 4. The enhanced State 6→4 transitions in ASD involved a shift from the somatomotor network (State 6) to the dorsal attention network (State 4). This aligns with literature on altered sensorimotor-cognitive coordination in ASD, potentially reflecting a dominance of bottom-up processing over top-down modulation (Amso et al., 2014; Maekawa et al., 2011). Such rapid switching between executive control states may indicate reduced attentional stability,

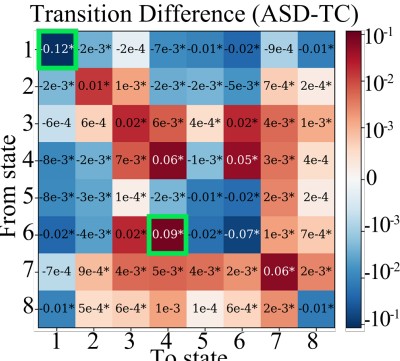

Figure 3: Group differences in state transitions for ABIDE-I, with green boxes highlighting the most discriminative patterns (*: $p < 0.001$).

consistent with prior findings (Ursino et al., 2022; Greenaway & Plaisted, 2005). In contrast, the TC group showed a greater tendency to remain in State 1, which is characterized by default mode, somatomotor, and visual networks. Additionally, complementary fractional dwell time analysis confirmed that the TC group exhibited significantly longer persistence in this state compared with the ASD group (see Appendix F.3.1). This pattern suggests more stable engagement of internally oriented cognition and sensory processing (He et al., 2018; Padmanabhan et al., 2017).

These findings reveal distinct brain network dynamics between diagnostic groups, demonstrating that the learned brain states capture clinically meaningful patterns consistent with established neurobiological alterations (Benkarim et al., 2021; Harikumar et al., 2021). Similar analyses for the ADHD-200 dataset are provided in Appendix F.3.2.

## 5.4 IN-DEPTH ANALYSIS OF CLUSTERING ASSIGNMENT

To validate the effectiveness of the orthonormal clustering mechanism, we analyzed the cluster assignment patterns compared to the learnable random initialization strategy (Random‡) described in Section 5.2. Specifically, we visualized the correspondence between the clusters and the 7 canonical functional networks defined by the Yeo atlas (Yeo et al., 2011), based on the soft assignment probability $\mathbf{P}$.

As illustrated in Figure 5, the orthonormal clustering (left) shows a relatively distinct alignment where specific clusters tend to map to corresponding functional networks. This suggests that the orthogonality constraint may facilitate the disentanglement of complex brain representations into

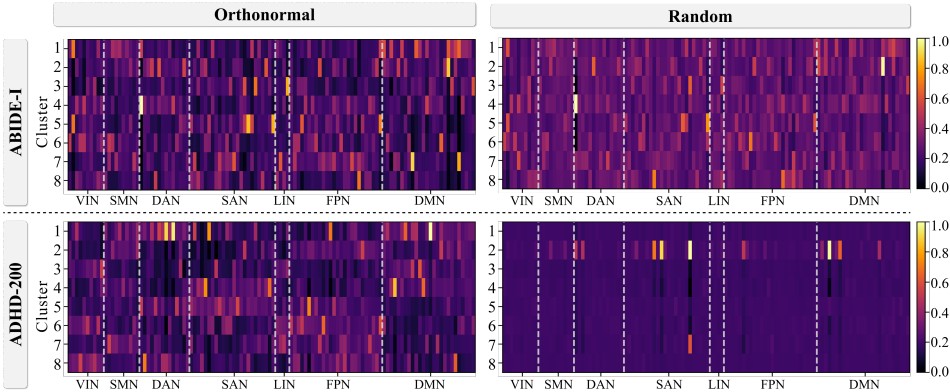

Figure 5: Visualization of cluster assignment probabilities mapped onto the 7 canonical functional networks, comparing the orthonormal (left) and random (right) initialization strategies. The values are globally normalized across all clusters for each dataset. (VIN: Visual, SMN: Somatomotor, DAN: Dorsal Attention, SAN: Salience/Ventral Attention, LIN: Limbic, FPN: Frontoparietal, DMN: Default Mode)

functionally coherent states, aligning with findings in prior work (Kan et al., 2022). In contrast, the random initialization (right) results in more distributed assignments, with some clusters exhibiting redundant patterns (particularly evident in the ADHD-200 dataset). These results indicate that the orthonormal constraint contributes to learning interpretable and non-redundant brain states.

## 6 CONCLUSION

In this work, we introduced DyBraSS, a novel structured SSM that jointly captures spatial dependencies and temporal dynamics for brain state modeling. Our approach leverages a global aggregation module that incorporates information from all brain regions into local ROI-level updates, thereby preserving the brain's network topology during state evolution. Furthermore, the integration of a clustering-based strategy within this module enables interpretable representations of dynamic brain states, facilitating both robust modeling and clinical interpretation. Comprehensive experiments on the ABIDE-I and ADHD-200 datasets demonstrate that our method consistently outperforms SOTA baselines and confirm the effectiveness of each component, including the orthonormal clustering mechanism Additionally, quantitative brain state analysis at both the individual- and group-levels reveal that the learned brain states align with known neurobiological alterations, providing valuable insights for computational neuroimaging and clinical applications. Looking ahead, the proposed model holds significant potential, and future work will extend it to a wider range of neuropsychiatric disorders and investigate methods for modeling more intricate brain states to further enhance generalizability and clinical utility.

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

## A    DETAILS OF DISCRETIZATION

We start from the continuous-time Dyn-SSM formulation for the $r$-th ROI:

$$\frac{d}{dt}\mathbf{h}_r(t) = \mathbf{A}_r\mathbf{h}_r(t) + \mathbf{B}_r\mathbf{x}_r(t) + \boldsymbol{\Phi}_r(t). \tag{17}$$

Consider the sampling interval $[t_k, t_{k+1}) = [k\Delta_r, (k+1)\Delta_r]$ where $\Delta_r$ is the discretization step for $r$-th ROI. Integrating Eq. 17 over this interval:

$$\int_{t_k}^{t_{k+1}} \frac{d}{dt}\mathbf{h}_r(t)\,dt = \int_{t_k}^{t_{k+1}} [\mathbf{A}_r\mathbf{h}_r(t) + \mathbf{B}_r\mathbf{x}_r(t) + \boldsymbol{\Phi}_r(t)]\,dt. \tag{18}$$

Although $\boldsymbol{\Phi}_r(t)$ depends on the entire hidden state $\{\mathbf{h}_i(t)\}_{i=1}^R$, we apply a modified ZOH approximation for computational tractability:

$$\mathbf{x}_r(\tau) \approx \mathbf{x}_{r,k+1}, \quad \boldsymbol{\Phi}_r(\tau) \approx \boldsymbol{\Phi}_{r,k+1}, \quad \tau \in [t_k, t_{k+1}). \tag{19}$$

To solve the integral, we apply the variation of constants formula with homogeneous solution $\mathbf{h}_r(t) = e^{\mathbf{A}_r(t-t_k)}\mathbf{h}_r(t_k)$, giving:

$$\mathbf{h}_r(t_{k+1}) = e^{\mathbf{A}_r\Delta_r}\mathbf{h}_r(t_k) + \int_{t_k}^{t_{k+1}} e^{\mathbf{A}_r(t_{k+1}-\tau)}[\mathbf{B}_r\mathbf{x}_r(\tau) + \boldsymbol{\Phi}_r(\tau)]\,d\tau \tag{20}$$

$$= e^{\mathbf{A}_r\Delta_r}\mathbf{h}_r(t_k) + \int_0^{\Delta_r} e^{\mathbf{A}_r s}\,ds \cdot [\mathbf{B}_r\mathbf{x}_{r,k+1} + \boldsymbol{\Phi}_{r,k+1}] \tag{21}$$

$$= e^{\mathbf{A}_r\Delta_r}\mathbf{h}_r(t_k) + \mathbf{A}_r^{-1}\left(e^{\mathbf{A}_r\Delta_r} - \mathbf{I}\right)[\mathbf{B}_r\mathbf{x}_{r,k+1} + \boldsymbol{\Phi}_{r,k+1}]. \tag{22}$$

Therefore, the discretized update equation becomes:

$$\mathbf{h}_{r,k+1} = e^{\mathbf{A}_r\Delta_r}\,\mathbf{h}_{r,k} + \mathbf{A}_r^{-1}\left(e^{\mathbf{A}_r\Delta_r} - \mathbf{I}\right)\mathbf{B}_r\,\mathbf{x}_{r,k+1} + \mathbf{A}_r^{-1}\left(e^{\mathbf{A}_r\Delta_r} - \mathbf{I}\right)\boldsymbol{\Phi}_{r,k+1}. \tag{23}$$

Defining the discretized parameters:

$$\bar{\mathbf{A}}_r = e^{\mathbf{A}_r\Delta_r}, \tag{24}$$

$$\bar{\mathbf{B}}_r = \mathbf{A}_r^{-1}(\bar{\mathbf{A}}_r - \mathbf{I})\,\mathbf{B}_r, \tag{25}$$

$$\bar{\boldsymbol{\Phi}}_{r,k+1} = \mathbf{A}_r^{-1}(\bar{\mathbf{A}}_r - \mathbf{I})\,\boldsymbol{\Phi}_{r,k+1}, \tag{26}$$

we obtain the final discretized Dyn-SSM:

$$\mathbf{h}_{r,k+1} = \bar{\mathbf{A}}_r\mathbf{h}_{r,k} + \bar{\mathbf{B}}_r\mathbf{x}_{r,k+1} + \bar{\boldsymbol{\Phi}}_{r,k+1}. \tag{27}$$

$$\tag{28}$$

## B    STABILITY OF CLOSED-LOOP SYSTEM

Given that the local dynamics are parameterized to be *asymptotically stable* following the S4D-Real parameterization based on HiPPO theory (Gu et al., 2020; 2022a), we focus here on demonstrating that the global feedback term, $\boldsymbol{\Phi}_{r,t}$ (which pertains to the $r$-th ROI at time $t$), acts as a bounded operator, thereby preserving the stability of the overall system.

The feedback term, which uses $K$ cluster-level global embeddings to redistribute information back to the ROI level based on Eqs. 13–15, is defined as:

$$\boldsymbol{\Phi}_{r,t} = \mathbf{P}_{r,t}\mathbf{G}_{r,t} \tag{29}$$

$$= \sum_{k=1}^K \pi_{r,k,t}\left(\sum_{j=1}^R \pi_{j,k,t}\mathbf{h}_{j,t-1}\right). \tag{30}$$

We note that the assignment weights satisfy $\sum_k \pi_{r,k,t} = 1$ and $\pi_{r,k,t} \geq 0 \; \forall r, k, t$. To verify the boundedness, let $\| \cdot \|$ denote the Euclidean norm. The magnitude of the feedback term satisfies the following inequality:

$$\|\mathbf{\Phi}_{r,t}\| = \left\| \sum_{k=1}^{K} \pi_{r,k,t} \left( \sum_{j=1}^{R} \pi_{j,k,t} \mathbf{h}_{j,t-1} \right) \right\| \tag{31}$$

$$\leq \sum_{k=1}^{K} \pi_{r,k,t} \sum_{j=1}^{R} \pi_{j,k,t} \|\mathbf{h}_{j,t-1}\| \qquad \text{(Triangle Inequality, since } \pi \geq 0\text{)} \tag{32}$$

$$\leq \sum_{k=1}^{K} \pi_{r,k,t} \sum_{j=1}^{R} \pi_{j,k,t} \left( \max_j \|\mathbf{h}_{j,t-1}\| \right) \qquad \text{(Bounding } \|\mathbf{h}_{j,t-1}\| \leq \max_j \|\mathbf{h}_{j,t-1}\|\text{)} \tag{33}$$

$$\leq \sum_{k=1}^{K} \pi_{r,k,t} R \left( \max_j \|\mathbf{h}_{j,t-1}\| \right) \qquad \text{(Bounding } \sum_j \pi_{j,k,t} \leq R\text{)} \tag{34}$$

$$= R \max_j \|\mathbf{h}_{j,t-1}\|. \qquad \text{(Since } \sum_k \pi_{r,k,t} = 1\text{)} \tag{35}$$

$$\tag{36}$$

This inequality shows that the global aggregation is a Lipschitz continuous mapping with respect to the hidden states, with Lipschitz constant at most $R$, which depends only on the number of ROIs. Combined with the asymptotically stable local dynamics, this implies that the feedback term acts as a bounded, Lipschitz perturbation of the local system, suggesting that the resulting closed-loop dynamics are resistant to unbounded growth of the states in practice.

## C  ALGORITHM

---

**Algorithm 1**: Forward Propagation of DyBraSS

---

**Input:**  Sequence of FC matrices $\mathbf{E} = \{\mathbf{e}_t\}_{t=1}^{T}$, Clusters $K$, State dim $N$
**Output:**  Global embeddings $\{\mathbf{G}_t\}_{t=1}^{T}$, Predicted FC sequence $\tilde{\mathbf{E}}'$
 1: Initialize orthonormal cluster centers $\mathbf{K} \in \mathbb{R}^{K \times N}$ via Gram-Schmidt            *// Eq. 12*
 2: Initialize hidden state $\mathbf{h}_0 \leftarrow \mathbf{0}$
 3: $\tilde{\mathbf{E}} \leftarrow \text{Embed}(\mathbf{E})$            *// Eq. 5*
 4: $\mathbf{x}, \mathbf{z} \leftarrow \text{Linear}(\tilde{\mathbf{E}})$            *// x-branch, z-branch generation*
 5: **for** $t = 1$ to $T$ **do**
 6:     // Global Aggregation (based on $\mathbf{h}_{t-1}$)
 7:     $\mathbf{P}_t \leftarrow \text{Softmax}(\langle \mathbf{h}_{t-1}, \mathbf{K} \rangle)$            *// Eq. 13: Soft assignment*
 8:     $\mathbf{G}_t \leftarrow \mathbf{P}_t^\top \mathbf{h}_{t-1}; \quad \mathbf{\Phi}_t \leftarrow \mathbf{P}_t \mathbf{G}_t$            *// Eq. 14, 15: Global context & Feedback*
 9:     // Dyn-SSM Update
10:     $\Delta_t, \mathbf{B}_t, \mathbf{C}_t \leftarrow \text{Linear}(\mathbf{x}_t)$            *// Eq. 11: Input-dependent params*
11:     $\bar{\mathbf{A}}_t, \bar{\mathbf{B}}_t, \bar{\mathbf{\Phi}}_t \leftarrow \text{Discretize}(\mathbf{A}, \mathbf{B}, \mathbf{\Phi}_t, \Delta_t)$            *// Eq. 10: ZOH discretization*
12:     $\mathbf{h}_t \leftarrow \bar{\mathbf{A}}_t \mathbf{h}_{t-1} + \bar{\mathbf{B}}_t \mathbf{x}_t + \bar{\mathbf{\Phi}}_t$            *// Eq. 8: State update*
13:     $\mathbf{y}_t \leftarrow \mathbf{C}_t \mathbf{h}_t + \mathbf{D} \odot \mathbf{x}_t$            *// Eq. 9: Output projection*
14: **end for**
15: $\mathbf{y} \leftarrow \{\mathbf{y}_t\}_{t=1}^{T}$
16: $\tilde{\mathbf{E}}' \leftarrow \mathbf{y} \odot \text{SiLU}(\mathbf{z})$            *// Gated output*

---

## D  DETAILS OF DATASETS

The ABIDE-I dataset consists of 1,112 individuals collected from 17 sites, comprising 539 participants with ASD and 573 TC. The ADHD-200 dataset includes 973 individuals: 362 ADHD-related participants (encompassing ADHD-combined, ADHD-hyperactive/impulsive, and ADHD-inattentive subtypes), 585 TD individuals, and 26 undefined cases. We excluded the undefined cases and merged all ADHD subtypes into a single ADHD category.

All fMRI data were preprocessed with Configurable Pipeline for the Analysis of Connectome (C-PAC). The pipeline included slice-timing correction, motion correction, distortion correction, coregistration to T1-weighted anatomy, normalization to the MNI152 template (final functional outputs at 3 mm), and spatial smoothing with a 4-mm full-width at half-maximum (FWHM) Gaussian kernel. To mitigate physiological and motion-related confounds, we applied nuisance regression in native space using Friston-24 motion terms, aCompCor with five principal components from WM/CSF (Behzadi et al., 2007), global signal regression, and linear and quadratic trends. ICA-Automatic Removal of Motion Artifacts (AROMA) was also applied. Temporal band-pass filtering (0.01–0.1 Hz) was performed after nuisance regression. Participants with preprocessing failures and severe head motion were excluded.

To compute dFC, we used a sliding-window approach (Allen et al., 2014) with a target window size of 15 s and a stride of 3 s. As described in Section 4.1, the window length and stride were normalized according to the site-specific TR to ensure temporal consistency across datasets. Within each window, Pearson correlation coefficients were calculated to obtain FC matrices. Only subjects with at least 100 valid windows were retained. After this procedure, 678 subjects remained for ABIDE-I (290 ASD, 388 TC) and 643 subjects for ADHD-200 (270 ADHD, 373 TD). Both datasets were parcellated into 114 ROIs using the Yeo 2011 atlas (Yeo et al., 2011). All baseline and comparison models used the same filtered subjects.

# E EXPERIMENTAL DETAILS

## E.1 IMPLEMENTATION

For both the ABIDE-I and ADHD-200 datasets, we used two stacked DynBrain-Mamba blocks with model dimension $D = 64$, Dyn-SSM state dimension $N = 16$, and a convolution kernel width of 4. The global aggregation module used 32 and 8 clusters in the first and second blocks, respectively. These cluster centers were initialized as orthonormal bases and remained fixed throughout training. The hidden dimensions for the input embedding MLP and the next-time FC prediction MLP were set to 512 for ABIDE-I and 256 for ADHD-200, while the classifier's convolution blocks used a hidden dimension of 8. The transition matrix $\mathbf{A}$ was initialized with negative real eigenvalues following HiPPO theory (Gu et al., 2020; 2022a), and the timescale parameter $\epsilon$ was initialized uniformly in $[0.001, 0.1]$. We employed SiLU (Elfwing et al., 2018) as the activation function throughout the model. Training was performed using the Adam optimizer with initial learning rates of 1e-4 for ABIDE-I and 5e-4 for ADHD-200, $l_2$ regularization with a weight decay of 1e-5, a batch size of 8 for both datasets, and a multitask objective with $\alpha = 0.5$. Our method was implemented in PyTorch, and all models, including ours and the baselines, were trained on an NVIDIA GeForce RTX 2080 GPU under Ubuntu 18.04.

## E.2 BASELINE PARAMETER TUNING

For all baseline models, we used the official open-source implementations and conducted hyperparameter tuning as described below, with all other settings following the original implementations.

**BrainNetCNN** (Kawahara et al., 2017): We tuned the number of hidden convolutional filters in E2E blocks {16, 32}, E2N filters {32, 64}, N2G filters {128, 256}, learning rates {1e-4, 1e-3, 1e-2}, weight decay {1e-5, 1e-4}, and batch sizes {16, 32, 64}. We selected the following configurations for both datasets: 32 hidden convolutional filters for E2E blocks, 64 filters for E2N layers, 256 filters for N2G layers, and a batch size of 16. The learning rate was set to 1e-4 for ABIDE-I and 1e-5 for ADHD-200.

**BrainNetTF** (Kan et al., 2022): We evaluated the output node number {32, 64, 100}, hidden size {256, 512, 1024}, learning rates {5e-5, 1e-4, 5e-4}, weight decay {1e-5, 1e-4, 1e-3}, and batch sizes {16, 32, 64}. After validation-based selection, we used 32 output nodes, a hidden size of 1024, a learning rate of 5e-5, and a weight decay of 1e-4 for ABIDE-I. For ADHD-200, we selected 32 output nodes, a hidden size of 512, a learning rate of 5e-5, and a weight decay of 1e-3. Both datasets used a batch size of 16.

**BolT** (Bedel et al., 2023): We varied window sizes {20, 40, 60}, learning rates {5e-5, 1e-4, 2e-4, 5e-4}, weight decay {0, 1e-5, 1e-4}, and batch sizes {16, 32}. After validation-based selection, we

selected window size of 20, learning rate of 2e-4, no weight decay, and batch size of 16 for both datasets.

**ContrastPool** (Xu et al., 2024): We evaluated hidden dimensions $\{64, 86, 128, 256\}$, learning rates $\{1e\text{-}3, 5e\text{-}3, 1e\text{-}2\}$, and batch sizes $\{10, 20, 40\}$. The final configurations employed a learning rate of 1e-2 for both datasets, with a hidden dimension of 86 and batch size of 20 for ABIDE-I, and a hidden dimension of 512 and batch size of 10 for ADHD-200.

**fMRI-S4** (El-Gazzar et al., 2022): We examined the number of convolutional layers $\{1, 2\}$, model dimensions $\{64, 128, 256\}$, state dimensions $\{32, 64, 128\}$, learning rates $\{1e\text{-}4, 5e\text{-}4, 1e\text{-}3\}$, weight decay $\{1e\text{-}5, 1e\text{-}4\}$, and batch sizes $\{16, 32, 64, 128\}$. We used 1 convolutional layer, model dimension of 256, learning rate of 1e-3, weight decay of 1e-5, and batch size of 128 for both datasets. The state dimension was set to 128 for ABIDE-I and 64 for ADHD-200.

**FST-Mamba** (Wei et al., 2025): We examined hierarchical depths $\{[2,2,2,2], [2,2,4,2], [2,2,6,2]\}$ representing connectivity and temporal encoder layers, embedding dimensions $\{16, 24\}$, learning rates $\{1e\text{-}4, 5e\text{-}4, 1e\text{-}3\}$, weight decay $\{1e\text{-}5, 1e\text{-}4\}$, and batch sizes $\{16, 32\}$. The final configuration for both datasets was hierarchical depth of [2,2,2,2], embedding dimension of 16, learning rate of 1e-3, weight decay of 1e-5, and batch size of 16.

**BrainMAP** (Wang et al., 2025): We assessed hidden dimensions $\{32, 64, 96, 128\}$, learning rates $\{5e\text{-}5, 1e\text{-}4, 1e\text{-}3, 5e\text{-}2\}$, weight decay $\{1e\text{-}5, 1e\text{-}4, 1e\text{-}3\}$, and batch sizes $\{16, 32\}$. The final configurations were: ABIDE-I with hidden dimension 32, learning rate 5e-2, and weight decay 1e-3; ADHD-200 with hidden dimension 64, learning rate 5e-2, and weight decay 1e-4. Both used batch size 16.

# F    ADDITIONAL BRAIN ANALYSIS RESULTS

## F.1    ANALYSIS DETAILS

For our brain state analysis, we utilized the results from all test subjects across all folds to compute group-level averages, ensuring a comprehensive evaluation. To determine the network configuration of each brain state, we used the top-1 state sequence derived from importance scores. For each state, we computed the average assignment probabilities across all time points where that particular state was identified as the top-1 state. Brain visualizations were generated using BrainNet-Viewer (Xia et al., 2013) (http://www.nitrc.org/projects/bnv/). Following previous studies (Mash et al., 2019; Yao et al., 2016), fractional dwell time was calculated as the proportion of total time that each subject spent in each state consecutively. We computed the average fractional dwell time for each state per subject, then aggregated these values at the group-level by averaging across subjects within each diagnostic group. To assess statistically significant differences in dwell times between groups, we performed permutation testing with 10,000 iterations.

## F.2    ADDITIONAL INDIVIDUAL-LEVEL ANALYSIS RESULTS

Representative cases for each dataset are shown in Figs. 6 and 7. In Fig. 6 for the ABIDE-I, the ASD group exhibits frequent 6→4 state transitions with elevated importance scores at these transition moments, whereas the TC group shows increased importance scores when remaining in State 1. Both patterns align with our group-level transition analysis in Fig. 8. Similarly, in Fig. 7 for the ADHD-200, the ADHD group shows a tendency to remain in State 8 with corresponding elevations in importance scores, whereas the TD group shows a preference for State 2 with elevated scores during these periods, consistent with the group-level findings in Fig. 11. These visualizations demonstrate that our model successfully captures individual-level dynamics while learning group-specific patterns that distinguish between diagnostic categories.

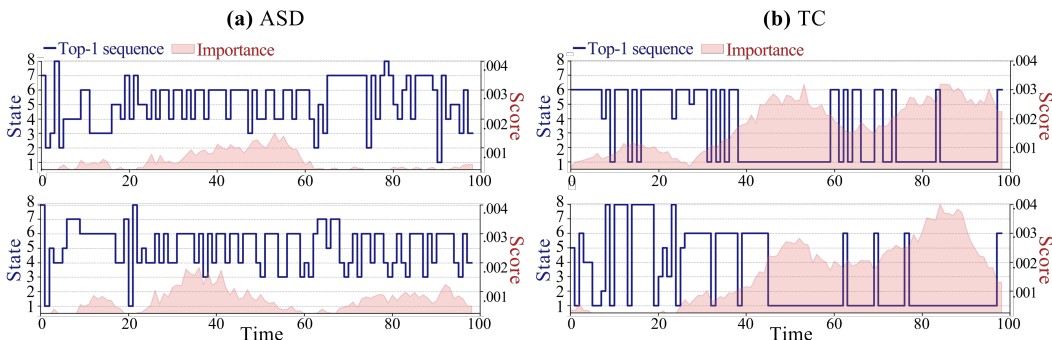

Figure 6: Brain state evolution with top-1 cluster sequence (blue) and importance scores (pink) for ASD and TC cases from ABIDE-I.

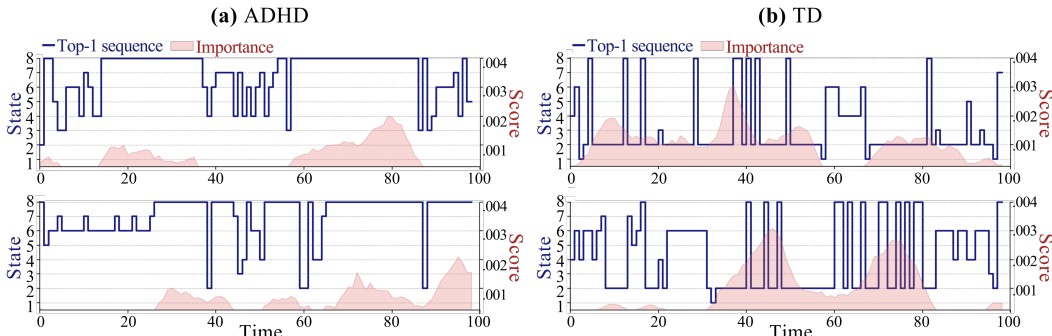

Figure 7: Brain state evolution with top-1 cluster sequence (blue) and importance scores (pink) for ADHD and TD cases from ADHD-200.

### F.3 ADDITIONAL GROUP-LEVEL ANALYSIS RESULTS

This section presents detailed group-level analysis results for each dataset. For ABIDE-I (Figs. 8–10) and ADHD-200 (Figs. 11–13), we report: (i) transition matrices, (ii) fractional dwell time analysis, and (iii) brain network configurations of each state. Detailed interpretations of the ABIDE-I results are provided in Section 5.3, while the ADHD-200 findings are discussed in Appendix F.3.2.

#### F.3.1 ABIDE-I GROUP COMPARISON

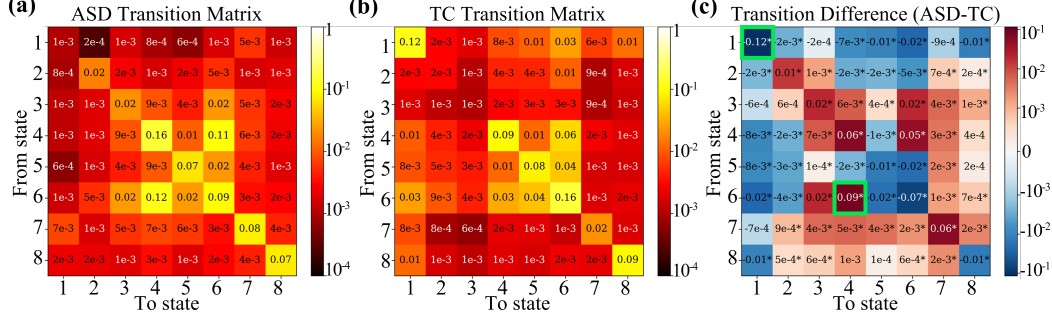

Figure 8: Transition matrices for (a) ASD group, (b) TC group, and (c) group difference (ASD - TC) in ABIDE-I dataset.

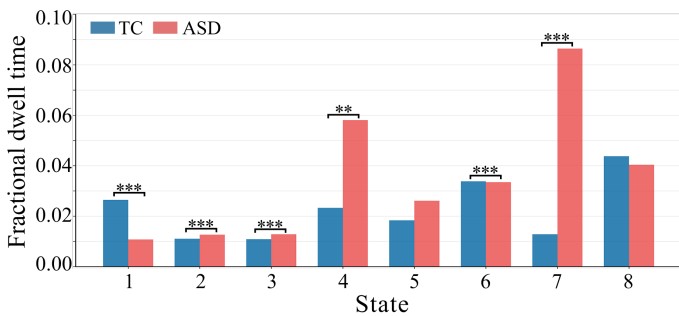

Figure 9: Fractional dwell time comparison between ASD and TC groups in ABIDE-I dataset. Statistical significance indicated by * $p < 0.05$, ** $p < 0.01$, *** $p < 0.001$.

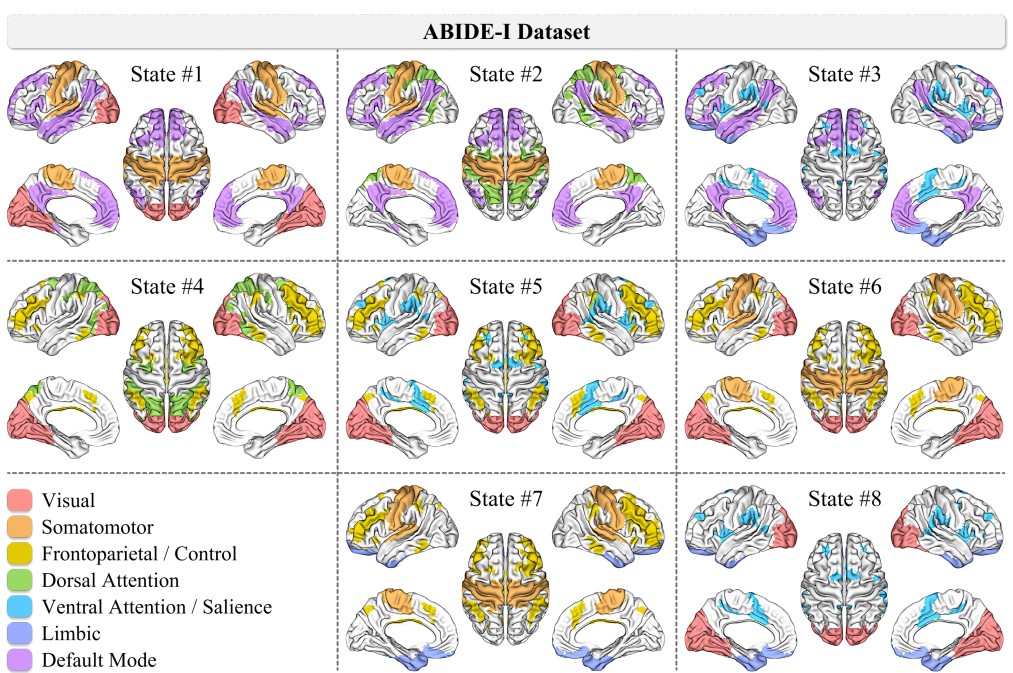

Figure 10: Brain network configurations of each state in ABIDE-I dataset.

### F.3.2 ADHD-200 GROUP COMPARISON

Examining the group difference transition matrix in Fig. 11 (c) and the brain network configurations of each state in Fig. 13, the ADHD group exhibits a pronounced tendency to remain in State 8, which involves visual, dorsal attention, and frontoparietal control networks. This pattern suggests sustained engagement of externally oriented attentional systems in ADHD, reflecting a bias toward stimulus-driven sensory processing with compromised executive-driven regulatory control (Mowinckel et al., 2017; Sidlauskaite et al., 2016). Consistently, the prolonged maintenance of this attention-control state in ADHD indicates reduced flexibility in transitioning between cognitive control configurations, aligning with findings of executive control deficits (Gao et al., 2025; Cortese et al., 2012).

In contrast, the TD group shows a greater tendency to remain in State 2, characterized by integrated frontoparietal control, somatomotor, and default mode network engagement. Complementary dwell time analysis confirms that the TD group exhibits significantly longer persistence in this state compared with the ADHD group (Fig. 12). This pattern reflects more stable coordination between executive control and internally oriented processing systems, consistent with robust intrinsic network integration observed in typical development (Tegelbeckers et al., 2015; Chen et al., 2021).

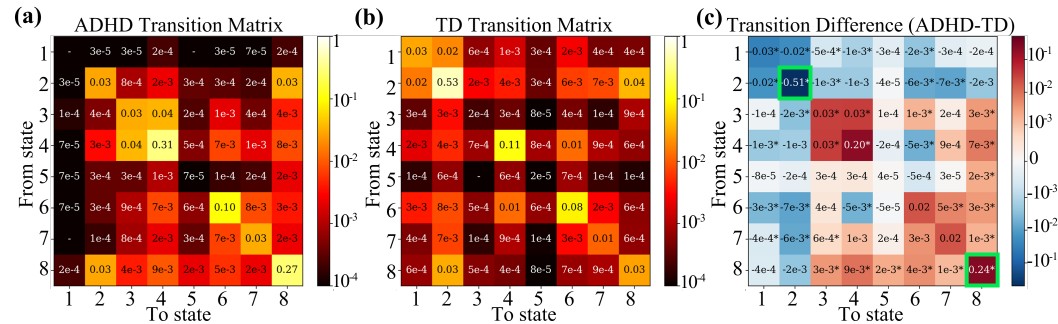

Figure 11: Transition matrices for (a) ADHD group, (b) TD group, and (c) group difference (ADHD - TD) in ADHD-200 dataset.

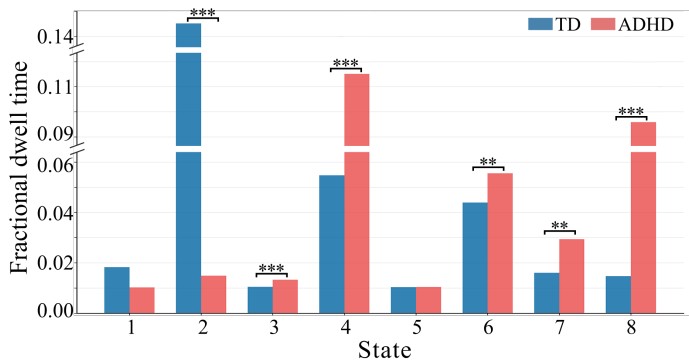

Figure 12: Fractional dwell time comparison between ADHD and TD groups in ADHD-200 dataset. Statistical significance indicated by $*\ p < 0.05$, $**\ p < 0.01$, $***\ p < 0.001$.

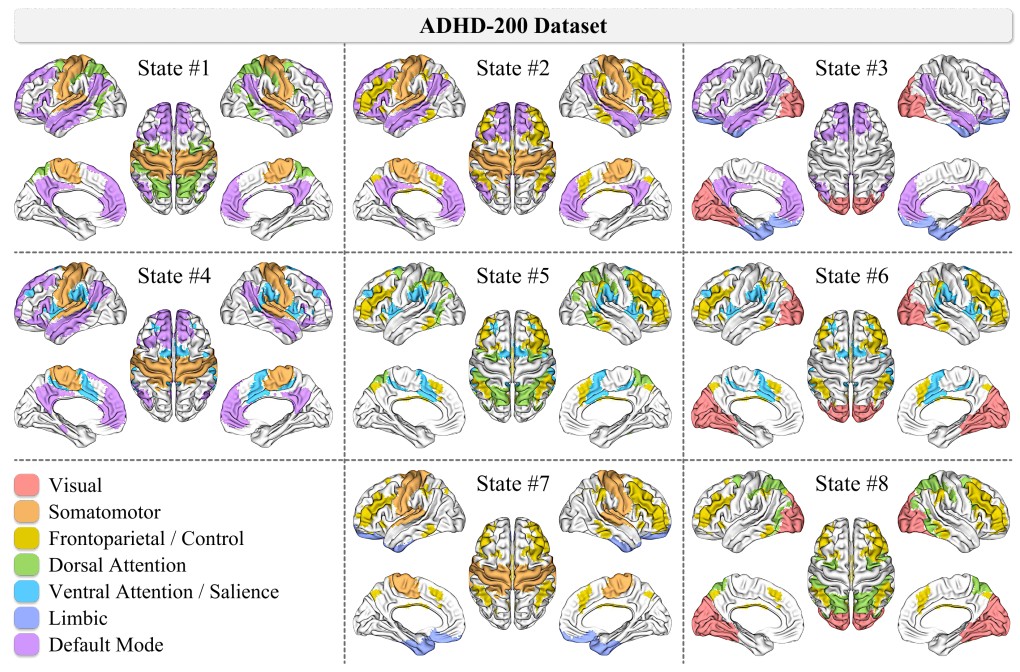

Figure 13: Brain network configurations of each state in ADHD-200 dataset.

## G  ADDITIONAL EXPERIMENT RESULTS

### G.1  COMPARISON RESULTS FOR AN ADDITIONAL DATASET

To evaluate the robustness and generalization capability of our proposed framework, we extended our experiments to the COBRE dataset (Mayer et al., 2013) for schizophrenia diagnosis, comprises 72 individuals with schizophrenia (SCZ) and 74 healthy controls (HC). We applied the same preprocessing pipeline as described for the ADHD-200 dataset in Appendix D, resulting in a final cohort of 145 subjects (74 HC and 71 SCZ).

The comparison results against baseline methods are presented in Table 4. As shown, our model consistently achieves superior performance compared to the baselines, demonstrating its effectiveness in diagnosing psychiatric conditions.

Table 4: Comparison with baseline models for classification on COBRE dataset.

| Method | COBRE | | | |
|---|---|---|---|---|
| | AUROC | ACC (%) | SEN (%) | SPC (%) |
| BrainNetCNN | $0.7010\pm0.06^*$ | $64.83\pm4.50^*$ | $66.10\pm18.65$ | $60.57\pm13.97$ |
| BrainNetTF | $0.7039\pm0.07^*$ | $64.14\pm5.23^*$ | $40.86\pm22.24^*$ | $\textbf{87.71}\pm\textbf{11.23}^*$ |
| BolT | $0.6915\pm0.12$ | $64.83\pm9.87^*$ | $70.38\pm11.85$ | $58.19\pm26.99^*$ |
| ContrastPool | $0.6602\pm0.16^*$ | $63.00\pm12.55^*$ | $58.67\pm21.81^*$ | $60.57\pm17.50^*$ |
| fMRI-S4 | $0.7042\pm0.09^*$ | $68.66\pm10.90$ | $53.71\pm21.01^*$ | $\underline{85.24}\pm10.87^*$ |
| FST-Mamba | $0.6919\pm0.08^*$ | $66.90\pm11.07^*$ | $59.33\pm13.68^*$ | $70.00\pm11.55$ |
| BrainMAP | $\underline{0.7190}\pm0.04$ | $\underline{69.17}\pm6.19^*$ | $68.95\pm4.33^*$ | $62.29\pm18.96$ |
| DyBraSS (Ours) | $\textbf{0.7369}\pm\textbf{0.03}$ | $\textbf{73.10}\pm\textbf{3.78}$ | $\textbf{74.77}\pm\textbf{4.17}$ | $71.52\pm10.26$ |

Best scores are highlighted in **bold**, second best are underline, and * indicates statistical significance ($p < 0.05$).

### G.2  ADDITIONAL ABLATION STUDY

**Influence of dFC variants.**  To assess the sensitivity of our model to the sliding-window hyperparameters used for dFC construction, we evaluated the performance across various window lengths ($w_{\mathrm{sec}}$) and stride sizes ($s_{\mathrm{sec}}$). As shown in Table 5 and Table 6, we observed a general trend where performance tends to decrease as the window length or stride increases, with the configuration of $w_{\mathrm{sec}} = 15$ and $s_{\mathrm{sec}} = 3$ yielding the best performance in our empirical evaluation.

Table 5: Ablation study on dFC window length $w_{\mathrm{sec}}$.

| $w_{sec}$ | ABIDE-I | | | | ADHD-200 | | | |
|---|---|---|---|---|---|---|---|---|
| | AUROC | ACC (%) | SEN (%) | SPC (%) | AUROC | ACC (%) | SEN (%) | SPC (%) |
| 10 | $0.6578\pm0.02^*$ | $65.31\pm1.97^*$ | $48.03\pm10.05^*$ | $60.48\pm8.14$ | $0.6485\pm0.03$ | $64.23\pm1.80$ | $55.01\pm10.29$ | $55.65\pm14.39$ |
| 20 | $0.6449\pm0.03^*$ | $64.13\pm1.68^*$ | $53.29\pm4.42$ | $68.34\pm6.68$ | $0.6375\pm0.04^*$ | $63.75\pm2.72$ | $52.76\pm14.62$ | $\textbf{62.42}\pm\textbf{11.98}$ |
| 25 | $0.6318\pm0.04^*$ | $63.87\pm3.67^*$ | $51.59\pm16.23$ | $\textbf{70.73}\pm\textbf{12.31}$ | $0.6330\pm0.01^*$ | $63.29\pm2.96^*$ | $45.55\pm13.60^*$ | $61.78\pm9.73$ |
| 30 | $0.6317\pm0.02^*$ | $63.44\pm2.01^*$ | $52.12\pm14.37$ | $69.58\pm9.85$ | $0.6260\pm0.02^*$ | $62.13\pm1.72^*$ | $56.04\pm7.08$ | $55.69\pm4.59^*$ |
| 15 | $\textbf{0.6904}\pm\textbf{0.01}$ | $\textbf{67.99}\pm\textbf{2.53}$ | $\textbf{63.38}\pm\textbf{6.98}$ | $68.27\pm7.13$ | $\textbf{0.6727}\pm\textbf{0.01}$ | $\textbf{66.23}\pm\textbf{2.34}$ | $60.93\pm9.41$ | $\underline{62.24}\pm5.91$ |

Best scores are highlighted in **bold**, second best are underline, and * indicates statistical significance ($p < 0.05$).

Table 6: Ablation study on dFC stride size $s_{\mathrm{sec}}$.

| $s_{sec}$ | ABIDE-I | | | | ADHD-200 | | | |
|---|---|---|---|---|---|---|---|---|
| | AUROC | ACC (%) | SEN (%) | SPC (%) | AUROC | ACC (%) | SEN (%) | SPC (%) |
| 5 | $0.6460\pm0.03^*$ | $64.17\pm2.97^*$ | $59.31\pm5.34$ | $56.01\pm7.64$ | $0.6449\pm0.01$ | $63.60\pm2.04^*$ | $53.29\pm8.75$ | $60.86\pm7.94$ |
| 7 | $0.6347\pm0.01^*$ | $62.73\pm2.39^*$ | $50.16\pm5.94^*$ | $63.78\pm8.93$ | $0.6317\pm0.04^*$ | $61.75\pm1.76$ | $\textbf{63.14}\pm\textbf{9.27}^*$ | $56.96\pm8.66$ |
| 9 | $0.6134\pm0.02^*$ | $60.71\pm2.47^*$ | $51.77\pm9.61^*$ | $61.50\pm4.56^*$ | $0.6212\pm0.05^*$ | $60.86\pm4.32^*$ | $46.20\pm11.80$ | $\textbf{69.05}\pm\textbf{9.51}$ |
| 3 | $\textbf{0.6904}\pm\textbf{0.01}$ | $\textbf{67.99}\pm\textbf{2.53}$ | $\textbf{63.38}\pm\textbf{6.98}$ | $\textbf{68.27}\pm\textbf{7.13}$ | $\textbf{0.6727}\pm\textbf{0.01}$ | $\textbf{66.23}\pm\textbf{2.34}$ | $60.93\pm9.41$ | $62.24\pm5.91$ |

Best scores are highlighted in **bold**, second best are underline, and * indicates statistical significance ($p < 0.05$).

**Influence of different brain atlases.**  To evaluate the robustness of our framework across different network scales and parcellation schemes, we extended our experiments using the AAL atlas (Tzourio-Mazoyer et al., 2002) (116 ROIs) and the Schaefer atlas (Schaefer et al., 2018) (100 ROIs). The results, presented in Table 7, indicate that while our model maintains competitive performance across different atlases, the configuration using the Yeo atlas (Yeo et al., 2011) yielded the best results.

Table 7: Ablation study on different brain atlases.

| Atlas | $R$ | ABIDE-I | | | | ADHD-200 | | | |
|---|---|---|---|---|---|---|---|---|---|
| | | AUROC | ACC (%) | SEN (%) | SPC (%) | AUROC | ACC (%) | SEN (%) | SPC (%) |
| AAL | 116 | 0.6633±0.01* | 65.34±3.22 | 48.20±7.99* | 63.39±10.74* | 0.6682±0.01 | 65.31±2.54* | 59.82±8.34* | **68.53±13.10** |
| Schaefer | 100 | 0.6544±0.05 | 64.37±3.65* | 51.84±6.67* | 63.57±22.48 | 0.6472±0.01* | 63.94±1.96* | 52.76±14.08* | 66.57±15.78 |
| Yeo | 114 | **0.6904±0.01** | **67.99±2.53** | **63.38±6.98** | **68.27±7.13** | **0.6727±0.01** | **66.23±2.34** | **60.93±9.41** | 62.24±5.91 |

Best scores are highlighted in **bold**, second best are underline, and * indicates statistical significance ($p < 0.05$).

## G.3 COMPUTATIONAL EFFICIENCY ANALYSIS

To evaluate the practical efficiency of the proposed framework, we compared the number of parameters and inference speed against baseline methods.

Measurements were performed using the hyperparameter configurations that yielded the best performance for each model. For a fair comparison of inference throughput, the batch size was standardized to 8 across all models. Note that ContrastPool (Xu et al., 2024) was excluded from the throughput comparison due to architectural constraints that couple pooling dimensions with the initialization batch size, preventing flexible batching during inference.

As presented in Table 8, our model demonstrates high parameter efficiency, requiring only 0.72M parameters (#Params), which is significantly more compact than recent structured SSM baselines like FST-Mamba (Wei et al., 2025). Regarding inference speed (Thpt.), we observed a trade-off where our model operates at a lower throughput due to the computational overhead of the iterative global aggregation mechanism. However, this design effectively enables explicit brain state trajectory modeling and neurobiological interpretability. We plan to explore more efficient architectural variants in future work to address this computational trade-off.

Table 8: Computational comparison with baseline models.

| Method | #Params (M) | Thpt. (k/s) |
|---|---|---|
| BrainNetCNN | 0.55 | 2.55 |
| BrainNetTF | 1.65 | 5.39 |
| BolT | 1.46 | 0.22 |
| ContrastPool | 0.43 | - |
| fMRI-S4 | 0.48 | 1.53 |
| FST-Mamba | 33.69 | 0.15 |
| BrainMAP | 0.12 | 0.42 |
| DyBraSS (Ours) | 0.72 | 0.13 |

Thpt. is reported in k samples/s (scaled by $10^3$).

## H GENAI USAGE DISCLOSURE

The authors employed generative AI language models for proofreading and linguistic refinement of the manuscript. No content was generated by AI tools.

