# OpenReview forum: "DyBraSS: Dynamic Brain State Modeling with State-Space Model"
_ICLR.cc/2026/Conference — Submitted to ICLR 2026_

### Official Review · Reviewer_fFeQ · 2025-10-26

**Soundness:** 2
**Presentation:** 2
**Contribution:** 2
**Rating:** 2
**Confidence:** 4

**Summary:**

This paper introduces a novel structured state space model (DyBraSS), which jointly models the spatiotemporal dependencies of brain dynamics within a unified framework by incorporating a clustering-based global aggregation module. The work demonstrates significant algorithmic innovation in handling dynamic functional connectivity from fMRI data, with systematic and comprehensive experimental design, showcasing outstanding diagnostic performance and model interpretability across multiple benchmark datasets. However, the paper exhibits notable shortcomings in theoretical rigor, motivation for methodological choices, statistical validation of results, and clarity of graphical representations.

**Strengths:**

1. DyBraSS successfully integrates ROI-level temporal evolution modeling with global brain state clustering within a single framework, effectively addressing the limitation of treating spatial and temporal dynamics in isolation, as seen in prior methods.

2. The core design, combining a dynamic SSM (Dyn-SSM) with an orthonormal cluster-based global aggregation mechanism, is novel. It preserves the brain's network topology while enhancing both ROI-level modeling capacity and the interpretability of state transitions.

3. The model demonstrates superior performance against a wide range of SOTA baselines (CNNs, Transformers, structured SSMs) on the ABIDE-I and ADHD-200 datasets.

**Weaknesses:**

1. The identifiability of the learned state transition matrices (Ar​) and observation matrices (Cr​) is not discussed. Given the high noise and low temporal resolution of fMRI data, it remains unclear whether these parameters are uniquely determined or confounded by equivalent parameterizations, which casts doubt on their neuroscientific interpretability.

2. The model introduces a feedback loop from the global clustering module to the local SSMs (Φr(t)). However, no stability analysis of this closed-loop dynamical system is provided. It is crucial to demonstrate that this feedback does not lead to divergent hidden states or unstable oscillations.

3. The theoretical motivation for using orthonormal clustering is weak. Orthogonality does not equate to statistical independence or neuroscientific dissociability. A stronger theoretical or empirical justification for why this method is superior to other clustering strategies is required.

4. The rationale behind key design choices is insufficiently explained: 1) Why was Mamba chosen as the foundational framework over other sequence models like Transformers? A comparative justification based on the related work is needed. 2) What is the motivation for the two-stream design ("x-branch" and "z-branch"), and specifically, the gating mechanism in the 'z-branch'? Why was gating chosen over an attention mechanism? 3) The reasoning for not using separate parameters for each feature dimension (as mentioned in Section 4.2.1, contrasting with the standard formulation in Section 3) is not provided.

5. The methodological description is challenging to follow. The data flow and interaction between modules are not clearly articulated. Providing a structured algorithm box or high-level pseudocode is strongly recommended, with explicit statements of inputs and outputs for each core module.

6. Performance comparison tables (e.g., Table 1) report means and standard deviations but lack statistical significance tests (e.g., t-tests, ANOVA). This makes it impossible to confirm the reliability of the performance improvements. Similarly, differences in transition matrices and dwell times between groups are described qualitatively without quantitative statistical validation.

7. An explanation is needed for why the proposed model does not achieve the best Specificity (SPC) scores in Table 1.

8. The results in Table 3 regarding the impact of Lpred​ on different metrics (e.g., increase in SEN but potential decrease in SPC) require deeper analysis

9. A detailed analysis of why other aggregation/clustering methods in Table 2 (e.g., Mean, Sum, Attention) perform poorly is necessary.

10. The paper does not report the model's parameter count, training/inference time, or a computational efficiency comparison with baseline methods. This is critical for assessing the method's practicality.

11. The conclusions are primarily based on two public datasets. Including a third independent dataset or conducting cross-dataset generalization experiments would significantly strengthen the claims.

12. Figure 1 (Model Overview) is inadequate. It fails to clearly illustrate the end-to-end data flow, temporal direction, hierarchical information exchange between modules, and the location of components mentioned in ablations (e.g., TR processing, MLPs). A complete redesign is necessary for clarity.

13. The visualization of brain state differences (e.g., in Figures 3 and 4) could be improved with better color schemes and layout. Including a state transition diagram to visually represent the probabilistic transitions between different brain states would greatly enhance interpretability.

14. Figures used to discuss ablation studies should clearly indicate which components were removed

15. The reference list contains numerous arXiv preprints. The authors should verify if these have been peer-reviewed and published subsequently, and prioritize citing the peer-reviewed versions where available.

**Questions:**

1. The identifiability of the learned state transition matrices (Ar​) and observation matrices (Cr​) is not discussed. Given the high noise and low temporal resolution of fMRI data, it remains unclear whether these parameters are uniquely determined or confounded by equivalent parameterizations, which casts doubt on their neuroscientific interpretability.

2. The model introduces a feedback loop from the global clustering module to the local SSMs (Φr(t)). However, no stability analysis of this closed-loop dynamical system is provided. It is crucial to demonstrate that this feedback does not lead to divergent hidden states or unstable oscillations.

3. The theoretical motivation for using orthonormal clustering is weak. Orthogonality does not equate to statistical independence or neuroscientific dissociability. A stronger theoretical or empirical justification for why this method is superior to other clustering strategies is required.

4. The rationale behind key design choices is insufficiently explained: 1) Why was Mamba chosen as the foundational framework over other sequence models like Transformers? A comparative justification based on the related work is needed. 2) What is the motivation for the two-stream design ("x-branch" and "z-branch"), and specifically, the gating mechanism in the 'z-branch'? Why was gating chosen over an attention mechanism? 3) The reasoning for not using separate parameters for each feature dimension (as mentioned in Section 4.2.1, contrasting with the standard formulation in Section 3) is not provided.

5. The methodological description is challenging to follow. The data flow and interaction between modules are not clearly articulated. Providing a structured algorithm box or high-level pseudocode is strongly recommended, with explicit statements of inputs and outputs for each core module.

6. Performance comparison tables (e.g., Table 1) report means and standard deviations but lack statistical significance tests (e.g., t-tests, ANOVA). This makes it impossible to confirm the reliability of the performance improvements. Similarly, differences in transition matrices and dwell times between groups are described qualitatively without quantitative statistical validation.

7. An explanation is needed for why the proposed model does not achieve the best Specificity (SPC) scores in Table 1.

8. The results in Table 3 regarding the impact of Lpred​ on different metrics (e.g., increase in SEN but potential decrease in SPC) require deeper analysis

9. A detailed analysis of why other aggregation/clustering methods in Table 2 (e.g., Mean, Sum, Attention) perform poorly is necessary.

10. The paper does not report the model's parameter count, training/inference time, or a computational efficiency comparison with baseline methods. This is critical for assessing the method's practicality.

11. The conclusions are primarily based on two public datasets. Including a third independent dataset or conducting cross-dataset generalization experiments would significantly strengthen the claims.

12. Figure 1 (Model Overview) is inadequate. It fails to clearly illustrate the end-to-end data flow, temporal direction, hierarchical information exchange between modules, and the location of components mentioned in ablations (e.g., TR processing, MLPs). A complete redesign is necessary for clarity.

13. The visualization of brain state differences (e.g., in Figures 3 and 4) could be improved with better color schemes and layout. Including a state transition diagram to visually represent the probabilistic transitions between different brain states would greatly enhance interpretability.

14. Figures used to discuss ablation studies should clearly indicate which components were removed

15. The reference list contains numerous arXiv preprints. The authors should verify if these have been peer-reviewed and published subsequently, and prioritize citing the peer-reviewed versions where available.

---

> ### Author Response · Authors · 2025-12-03
>
> We thank the reviewer for the constructive feedback and address the weaknesses and questions as follows.
>
> ---
> ### **(W1, Q1) Identifiability and Interpretability of $\mathbf{A}_r$ and $\mathbf{C}_r$**
>
> We acknowledge the reviewer’s concern regarding the identifiability of the learned parameters $\mathbf{A}_r$ and $\mathbf{C}_r$, particularly given the noise characteristics of fMRI data. We clarify that our model mitigates the issue of equivalent parameterizations and ensures interpretability through structured constraints grounded in the S4 [1] and HiPPO [2] theory.
>
> **1. Structured Constraints for Identifiability**
> We address the issue of *equivalent parameterizations* (often referred to as rotational ambiguity) by adopting the structured **diagonal state space (S4D)** parameterization [3]. While standard dense SSMs are invariant to similarity transformations, constraining $\mathbf{A}_r$ to be **diagonal** restricts the search space to a canonical form, effectively resolving this ambiguity [3].
>
> In addition, we initialize $\mathbf{A}_r$ with the **HiPPO matrix**, which mathematically structures the latent state to approximate the input history using a specific polynomial basis (e.g., Legendre) [2]. Empirical studies in [3] indicate that maintaining this specific initialization structure is vital for performance, suggesting that the learned parameters represent stable, mathematically grounded operators rather than arbitrary values.
>
> In this framework, the observation matrix $\mathbf{C}_r$ acts as a readout projection operating on a latent basis that is structurally anchored by the diagonal form and initialization of $\mathbf{A}_r$. This structural constraint ensures that $\mathbf{C}_r$ is well-defined relative to the fixed basis, preventing it from being confounded by arbitrary transformations.
>
>
> **2. Scope of Neuroscientific Interpretability**
> Regarding the neuroscientific utility, it is important to clarify that we do not interpret the individual entries of $\mathbf{A}_r$ and $\mathbf{C}_r$ as direct neurophysiological quantities. Instead, these parameters define the dynamics that generate the **latent brain states**. Our neuroscientific interpretability is derived from the clustering of these latent states, which reveals recurring spatiotemporal patterns. As demonstrated in our results, these discovered states align well with known functional networks, confirming that the model captures meaningful biological signals through the identifiable dynamics established by the structured SSM parameters.
>
>
> ### References
> > [1] Gu et al., "Efficiently modeling long sequences with structured state spaces," ICLR, 2022.
> [2] Gu et al., “HiPPO: Recurrent memory with optimal polynomial projections,” NeurIPS, 2020.
> [3] Gu et al., “On the parameterization and initialization of diagonal state space models,” NeurIPS, 2022.

---

> ### Author Response · Authors · 2025-12-03
>
> ### **(W2, Q2) Stability of the Feedback Loop**
>
> We thank the reviewer for this constructive comment regarding the closed-loop stability.
>
> We would like to clarify that the proposed feedback loop is structurally designed to mitigate the risk of divergence.
> The global feedback term $\boldsymbol{\Phi}_{r,t}$ is computed via a softmax-based weighted aggregation of the current hidden states. Since the weights are nonnegative and normalized via Softmax, the feedback corresponds to a bounded weighted aggregation of the hidden states rather than an unbounded amplification source.
>
> Consequently, this feedback mechanism functions as a redistribution of contextual information. When this bounded feedback is integrated into the local SSMs, which are parameterized to be asymptotically stable using the HiPPO-based formulation mentioned above, the overall system remains well-posed and is robust against unstable oscillations. A detailed derivation of this property is provided in **Appendix B.**

---

> ### Author Response · Authors · 2025-12-03
>
> ### **(W3, Q3) Theoretical and Empirical Justification for Orthonormal Clustering**
>
> We appreciate the reviewer’s constructive comment regarding the distinction between orthogonality and statistical independence. We acknowledge that while orthogonality does not strictly guarantee independence in a general statistical sense, we employ it as a structural constraint to enhance discriminability within the soft clustering mechanism.
>
> From a theoretical perspective, the motivation for using orthonormal bases lies in maximizing the variance of cluster assignment probabilities, as analytically derived in [1]. By enforcing orthogonality among cluster centers, the model is mathematically constrained to maximize the distance between latent states in the feature space. This structural bias encourages the model to disentangle complex brain dynamics into functionally distinct modules, thereby reducing ambiguity in the assignment process and preventing mode collapse, where multiple clusters might otherwise represent redundant signals.
>
> To provide empirical justification, we conducted a comparative analysis visualizing the relationship between our orthonormal clusters and the canonical Yeo-7 functional networks in **Section 5.4**. As illustrated in **Figure 5**, the orthonormal clustering exhibits a clear alignment where specific clusters selectively map to distinct functional networks. In contrast, unconstrained (random) clustering results in distributed and overlapping assignments. This suggests that orthogonality serves as an effective structural constraint that guides the model to disentangle complex brain dynamics into functionally coherent and non-redundant states.
>
>
> ### Reference
> > [1] Kan et al., “Brain network Transformer,” NeurIPS, 2022.

---

> ### Author Response · Authors · 2025-12-03
>
> ### **(W4, Q4) Design Choices**
> We appreciate the reviewer’s request for clarification on our architectural design choices. We provide the rationale for each component below:
>
> **1. Choice of Mamba (SSM) over Transformers**
> We selected the Mamba framework primarily for its suitability for **continuous signal modeling**. Structured SSMs have recently emerged as a powerful paradigm for extending standard state-space models to deep learning, showing significant promise in time-series forecasting. Unlike Transformers, which treat time steps as discrete tokens, SSMs are rooted in continuous-time differential equations. This formulation offers a framework that naturally aligns with the continuous biological fluctuations inherent in BOLD signals (fMRI), facilitating the modeling of temporal evolution.
>
> **2. Two-stream Design and Gating Mechanism**
> The two-stream architecture (x-branch and z-branch) with multiplicative gating is adopted from the standard Mamba block design [1], functioning as a robust selection mechanism. The gating mechanism allows the model to selectively propagate information, thereby controlling the information flow through the network.
>
> Regarding the comparison with attention, while attention mechanisms also provide selectivity, they incur high computational costs due to global pairwise calculations. In contrast, the gating mechanism achieves input-dependent selectivity via efficient **element-wise operations** (following linear projections), which significantly lowers computational overhead. This approach has been proven both efficient and effective in prior studies [1].
>
> **3. Unified Parameterization across Feature Dimensions**
> Standard SSM implementations often apply independent SSMs to each feature dimension. However, given the multi-channel nature of fMRI data (many ROIs), applying independent dynamics to every feature dimension would significantly increase model size. Therefore, as described in **Section 4.2.1**, we employ a unified parameterization (using projection matrices $\mathbf{B}_r \in \mathbb{R}^{N \times D}$) to integrate the entire $D$-dimensional ROI embedding into the latent state. This design choice aims to achieve parameter efficiency while enabling selectivity at the ROI level, effectively preserving the distinct dynamics of each brain region.
>
> ### Reference
> > [1] Gu et al., "Mamba: Linear-time sequence modeling with selective state spaces," COLM, 2024.

---

> ### Author Response · Authors · 2025-12-03
>
> ### **(W5, Q5) Pseudo-code**
> We appreciate the reviewer’s suggestion to improve the clarity of our methodological description.
>
> In response, we have added a structured Algorithm box (**Algorithm 1**) in **Appendix C**. This pseudo-code explicitly details the input-output flow and the step-by-step interaction between the core modules, including the global aggregation mechanism and the Dyn-SSM updates. We believe this addition significantly clarifies the data flow and the overall training procedure of the proposed framework.
>
> ### **(W6, Q6) Statistical Validation**
> We appreciate the reviewer’s emphasis on statistical validation.
>
> To verify the reliability of the performance improvements, we conducted paired $t$-tests on the 5-fold cross-validation results and updated the comparative tables, including Table 1 to mark statistically significant improvements ($p < 0.05$) with an asterisk (*).
>
> Regarding the group-level analysis, we performed independent $t$-tests to assess differences in transition probabilities between groups, and significant transitions are now explicitly marked in the revised transition matrices. Additionally, we confirm that quantitative statistical validation for dwell times was already included in **Appendix F.3** (**Figures 9 and 12**), where significant group differences were identified using permutation testing.

---

> ### Author Response · Authors · 2025-12-03
>
> ### **(W7, Q7) Specificity Scores**
> We acknowledge that our model does not achieve the highest SPC compared to some baseline methods. We interpret this result in the context of the inherent trade-off between SEN and SPC.
>
> As observed in **Table 1**, baselines achieving the highest SPC often exhibit significantly lower SEN (e.g., fMRI-S4 on ABIDE-I: SPC $\approx$ 79\% vs. SEN $\approx$ 40\%), suggesting a bias toward predicting the negative class (healthy controls). In contrast, our model achieves the highest AUROC and ACC while maintaining a **balanced performance** between SEN and SPC (both $>$ 60\%). Given the clinical importance of minimizing false negatives, this indicates that our approach offers a robust diagnostic capability without the class bias observed in the baselines.
>
> ### **(W8, Q8) Impact of Prediction Loss**
> We interpret the trade-off observed in **Table 3** as a trade-off where the auxiliary prediction task ($\mathcal{L}_{\rm pred}$) substantially boosts SEN and overall performance (AUROC, ACC) at the cost of a slight decrease in SPC.
>
> The $\mathcal{L}_{\rm pred}$ objective explicitly models temporal dynamics at the individual level by predicting next-time connectivity. This capability enhances the model's sensitivity in detecting positive cases (patients), which is often prioritized in clinical diagnostic contexts. Therefore, despite the slight reduction in SPC, the consistent improvements in AUROC and ACC confirm that incorporating individual-level dynamic modeling contributes to a more robust and effective diagnostic framework.

---

> ### Author Response · Authors · 2025-12-03
>
> ### **(W9, Q9) Performance Analysis of Aggregation Methods**
> We appreciate the reviewer’s request for a detailed performance analysis. We interpret the performance differences as stemming from how well each aggregation mechanism aligns with the characteristics of dynamic brain activity.
>
> Regarding Mean and Sum aggregation, these methods rely on simple global pooling. We attribute their lower performance to the potential oversimplification of the brain's distinct functional modularity, which may limit the capture of fine-grained spatial information. Similarly, while Attention is flexible, it focuses on capturing dense pairwise relationships between all regions.
>
> In contrast, our proposed method structures ROI dynamics based on a compact set of orthonormal bases. This structural constraint forces the model to explicitly identify **recurring, distinct brain states**. This capability to disentangle complex dynamics into interpretable states appears to be the key factor driving the superior performance of our method compared to other strategies.
>
> ### **(W10, Q10) Computational Comparison**
> We appreciate the reviewer’s feedback regarding the practical aspects of our model. To address this, we have conducted a computational analysis comparing our method with baselines, and the full results are now included in **Appendix G.3**.
>
> Our analysis highlights that the proposed model is parameter-efficient, requiring only 0.72M parameters (on ABIDE-I). This is significantly more compact than recent structured SSM baselines such as FST-Mamba (33.69M) and comparable to lightweight CNN architectures.
>
> Regarding computational cost during inference, we observed that our model has lower throughput compared to some baselines. This is primarily due to the iterative global aggregation mechanism, which is essential for integrating spatial context into the temporal dynamics at each step. While this introduces a computational overhead, we believe the trade-off is justified, as it enables the explicit modeling of **brain state trajectories** and provides **neurobiological interpretability**. We acknowledge the importance of efficient implementations and plan to explore optimizations for the aggregation module in future work.
>
> ### **(W11, Q11) Additional Dataset**
> We appreciate the reviewer’s constructive suggestion to strengthen our claims by including an independent dataset.
> In response, we expanded our evaluation to a third dataset, **COBRE** [1] (schizophrenia), as detailed in **Appendix G.1**. While our initial focus was on neurodevelopmental disorders, this addition serves to verify the model's generalization capability on a distinct psychiatric condition.
>
> As demonstrated in the new results, our model consistently achieved the best performance in AUROC, ACC, and SEN compared to the baselines. Notably, the improvement in ACC was statistically significant ($p < 0.05$) against most competing methods, demonstrating the effectiveness of our framework on the independent schizophrenia dataset. We acknowledge the importance of cross-dataset generalization and plan to extend our analysis to a wider spectrum of disorders in future work.
>
> ### Reference
> > [1] Mayer et al., "Functional imaging of the hemodynamic sensory gating response in schizophrenia," Hum Brain Mapp., 2012.

---

> ### Author Response · Authors · 2025-12-03
>
> ### **(W12–W14, Q12–Q14) Enhancement of Visual Illustrations**
> We appreciate the reviewer’s suggestions to improve the clarity of our visual illustrations. We have updated the figures in the revised manuscript as follows:
> 1.  **Model overview (Figure 1):** We have refined the figure to explicitly illustrate key components such as TR normalization, which clarifies the structural context of the components discussed in the ablation studies.
> 2.  **Brain state visualization:** For the brain state analysis (e.g., Figures 3), we refined the layout by incorporating probabilistic transition values directly into the transition diagrams. This visually represents the dynamics between brain states more clearly, thereby enhancing interpretability.
>
> ### **(W15, Q15) Reference Updates**
> We appreciate the reviewer’s feedback regarding the reference list. Following the suggestion, we have thoroughly verified the publication status of the cited preprints and updated the bibliography to prioritize peer-reviewed versions wherever available.

---

### Official Review · Reviewer_zr9w · 2025-10-28

**Soundness:** 2
**Presentation:** 3
**Contribution:** 2
**Rating:** 6
**Confidence:** 3

**Summary:**

The authors propose DyBraSS, which combines a structured state-space model with orthonormal cluster aggregation to simultaneously model temporal dynamics and global spatial context at the ROI level. This approach allows for interpretable modeling of dynamic brain states from rs-fMRI and is used for disease classification (ASD, ADHD). The authors compare their results against multiple state-of-the-art baselines on ABIDE-I and ADHD-200, reporting robust performance improvements and providing individual and group-level brain state analysis.

**Strengths:**

1. Combining ROI-wise SSM with "soft assignment to orthogonal clusters (brain-state) → aggregation → feedback to ROI" forms a closed-loop spatiotemporal coupling mechanism, which is conceptually natural and explainable.
2.  On ABIDE-I and ADHD-200, DyBraSS generally outperforms various representative baselines  in AUROC/ACC/SEN. Table 1 shows the specific improvements .
3. The authors examine the effects of various global aggregation strategies, as well as the next-time FC auxiliary loss and TR regularization, providing intuitive comparisons.
4. Provides individual/group-level state transition differences, dwell time analysis, and brain network visualization, attempting to link model findings with neurobiological literature and provide clinical interpretability.

**Weaknesses:**

1.Table 1 reports mean ± standard deviation, but does not explicitly report significance tests for baseline vs. DyBraSS. Given that the improvements in AUROC are mostly ~0.02–0.04, statistical tests demonstrating that the improvements are not due to chance are needed, along with detailed per-fold values ​​or boxplots. Please also provide additional significance test results.
2.Appendix D.1 states, "We choose to calculate group-level means under the optimal validation AUROC fold and analyze only subjects correctly classified by the model." Taking a group average only for correctly classified subjects introduces bias and may exaggerate the consistency of the model's interpretable analysis. Please provide additional analysis results for all subjects; or at least demonstrate consistency between the results for only correctly classified samples and the full sample.

**Questions:**

1.How are cluster centers (K) maintained/updated? Are they learned during training?The paper states, "K cluster centers are defined as an orthogonal basis and obtained from the initial random vector V by Gram-Schmidt" . However, it is unclear whether K is updated during training, or what the frequency of the Gram-Schmidt method is. This determines whether the cluster representation is a static basis or a learnable subspace that evolves with the data, which directly impacts the novelty and interpretability of the method.
2.The authors use sliding-window Pearson correlation to construct dFC. The choice of sliding window can significantly affect dFC representation. Please provide additional sensitivity analysis of window length/stride, or try a non-windowed approach to verify the robustness of the method.
3.The interpretive use of Captum to derive importance scores is excellent, but should avoid presenting only representative cases. Please provide additional statistical summaries and explain the consistency of importance scores with traditional neuroscience metrics.

---

> ### Author Response · Authors · 2025-12-03
>
> We thank the reviewer for the constructive feedback and address the weaknesses and questions as follows.
>
> ---
>
> ### **(Q1) Maintenance and Update Mechanism of Cluster Centers**
> We appreciate the reviewer’s query regarding the training dynamics of the cluster centers. To clarify, in our orthonormal clustering module, the cluster centers ($\mathbf{K}$) are initialized once via the Gram-Schmidt process and remain **fixed** throughout the training phase.
>
> This design choice is intentional to guarantee strict orthogonality [1], which would otherwise be compromised if the centers were updated via gradient descent. In addition, this approach ensures **stability and interpretability** by establishing a fixed orthonormal coordinate system. Acting as stable anchors, these bases guide the model to map dynamic ROI embeddings into a structured space, encouraging the learned representations to become discriminative and disentangled rather than drifting into redundant modes.
>
> To empirically validate the impact of this design on interpretability, we compared the assignment patterns of our learned orthonormal clusters against randomly initialized (learnable) clusters with respect to the Yeo-7 functional networks, as detailed in **Section 5.4**. As shown in **Figure 5**, the orthonormal clusters exhibit a clear alignment with distinct functional networks, whereas the random initialization results in noisy, indiscriminative patterns or assignments heavily skewed toward specific clusters. This confirms that the orthogonality constraint effectively facilitates discriminative modeling that potentially reflects functional coherence.
>
>
> ### Reference
> > [1] Kan et al., “Brain network Transformer,” NeurIPS, 2022.

---

> ### Author Response · Authors · 2025-12-03
>
> ### **(Q2) dFC Representation and Robustness**
> We appreciate the reviewer’s suggestion regarding the dFC construction.
>
> We conducted ablation studies on the sliding-window parameters, specifically varying the window length ($w_{\text{sec}}$) and stride ($s_{\text{sec}}$). The detailed results are presented in **Table 5-6 in Appendix G.2**.
>
> As observed in the results, the model's performance tends to decrease when the window length or stride is increased. We interpret this decline as an indication that larger temporal windows may over-smooth the data, making it difficult to capture the spontaneous brain state transitions essential for this task. Consequently, we selected the parameters ($w_{\text{sec}}=15, s_{\text{sec}}=3$) that empirically yielded the best performance to report our main results.

---

> ### Author Response · Authors · 2025-12-03
>
> ### **(Q3) Additional Statistical Analysis**
> We appreciate the reviewer’s constructive suggestion regarding the scope of our interpretability analysis.
>
> We clarify that the individual-level visualizations were initially presented to demonstrate the model's capability to capture fine-grained temporal dynamics at the subject level.
> However, we fully agree that establishing reliability requires statistical validation beyond representative cases. To address this, we emphasize that our analysis incorporates quantitative group-level metrics, specifically transition matrices and fractional dwell times, as detailed in **Section 5.3** and **Appendix F**.
>
> We performed statistical tests on these metrics, and the results reveal statistically significant group differences that align with established neurobiological findings. This confirms that the importance scores learned by our model are not merely artifacts of specific cases but reflect consistent, biologically meaningful patterns across the population.
>
> ### **(W1) Statistical Report**
> We thank the reviewer for highlighting the importance of rigorous statistical validation, especially given the performance margins.
>
> To demonstrate that the improvements are statistically significant, we conducted paired $t$-tests on the 5-fold cross-validation results, comparing DyBraSS with the baseline models.
>
> We have updated the comparative tables, including Table 1, to explicitly indicate statistical significance; specifically, results with $p < 0.05$ are marked with an asterisk (*). While the reported standard deviations already provide insight into the stability of the model across folds, these additional statistical tests further confirm that our method consistently outperforms the baselines across different data splits, verifying the reliability of the reported improvements.

---

> ### Author Response · Authors · 2025-12-03
>
> ### **(W2) Brain Analysis Protocol**
> We agree with the reviewer that restricting the analysis to correctly classified subjects may introduce selection bias.
> In response, we have revised the entire brain state analysis to include all test subjects across all folds, ensuring an unbiased evaluation. The updated results are presented in **Section 5.3** and **Appendices F**.
>
> Crucially, we confirmed that the core findings from the full sample are consistent with our initial findings. While we observed minor fluctuations in the values, the dominant brain state patterns and transition dynamics specific to each diagnostic group remained prominent. We further validated this consistency using independent two-sample $t$-tests on the transition maps (**Figure 3** and **Figure 11**), which demonstrated that the discriminative patterns between groups remain statistically significant ($p < 0.001$) even when analyzing the entire test set.

---

### Official Review · Reviewer_dXQf · 2025-10-31

**Soundness:** 2
**Presentation:** 3
**Contribution:** 2
**Rating:** 2
**Confidence:** 5

**Summary:**

This paper presents DyBraSS, a novel structured SSM that unifies spatial and temporal modeling within a single framework, enhancing ROI-level modeling capacity and interpretability through a clustering-based global aggregation module. Compared with various SOTA methods on ABIDE-I and ADHD-200, it showed a stable improvement in indicators such as AUROC/ACC. Brain state analysis at both the individual- and group-levels reveal that the learned brain states align with known neurobiological alterations, providing valuable insights for computational neuroimaging and clinical applications.

**Strengths:**

(1) The approach to the problem is natural and significant: combining spatial topology (information between ROIs) with the SSM to describe jointly models inter-ROI interactions during temporal state evolution, this perspective is highly compatible with understanding dFC and has potential neurointerpretability.

(2) The proposed leverages a global aggregation module that incorporates information from all brain regions into local ROI-level updates, thereby preserving the brain’s network topology during state evolution.

(3) The clear organization and presentation of this article make it a pleasure to read.

**Weaknesses:**

1. In Appendix D.1, the authors state that " we selected the model from the fold with the best validation AUROC score and analyzed only subjects that were correctly classified by this model to compute group-level averages" , this strategy introduces serious selection bias (only analyzing the samples "selected" by the model), which may exaggerate the robustness of the said group differences and interpretability conclusions.

2. The comparison between orthonormal cluster design and alternative solutions is shallow. Although Table 2 provides a comparison of multiple aggregation methods, it lacks a more in-depth analysis of why orthonormal can bring advantages (such as cluster center visualization or correspondence analysis with brain functional modules).

3. The explicit appearance of Eq.10 $A_r^{-1}$ imposes requirements on the eigenvalues/reversibility of matrix $A_r$. The paper does not discuss how to ensure numerical stability, whether instability (gradient explosion/disappearance) occurs during training, or how to constrain $A_r$.

4. Although DyBraSS performs well on two public datasets, these datasets may not fully cover all the variability of fMRI data. It is necessary to further verify the model's generalization ability on more diverse datasets to ensure its stability and reliability in practical applications. Such as, ADNI, OASIS, PPMI et al.

5. The comparison methods are somewhat limited, as many existing fMRI analysis approaches were not considered — for example, the original Mamba, Graphormer, NAGphormer, NeuroPath, NeuroGraph, and ContrastPool.

6. The model did not analyze different network scales, which means it did not explore the performance of the model under different brain atlases (such as AAL 116, Schaefer 1000).

**Questions:**

(1)	Why adopt the practice of analyzing only the subjects correctly classified by the model in the folds with the best validation AUROC scores to calculate the population average? This will lead interpretive analysis to favor samples that the model "agrees with", thereby exaggerating the differences. If this screening is removed (i.e., a group-level analysis is conducted on all test set samples or all folded test samples), will the results be consistent?

(2)	Why orthonormal bases via Gram-Schmidt (Eq. 12)?  How sensitive is the model to initial $V$ vectors?

(3)	The construction of orthonormal cluster defines orthogonal bases as cluster centers. In the "Orth(Ours)" mode (Table 2), does this orthogonalization remain fixed during training (initialized only once), or can the cluster center be updated? If it can be updated, how can orthogonality be guaranteed (or is it not mandatory)? If it cannot be updated, will it limit the presentation ability?

(4) 	What is the specific numerical range and stability of Eq.11 $\Delta_{r,t}$ (since $\epsilon_r \in R$ is a learnable parameter)? $\Delta_{r}$ determines the scale of $\overline{A}_r=\exp(\Delta_rA_r)$.

 Is there $\overline{A}$ situation where $\Delta_{r}$ is so large that $\overline A$ approaches singularity or explod? Should $\Delta_{r}$ be truncated or regularized?

(5) Tables 1 and 2 present the mean ±std, but do not explicitly state whether the differences among different methods are significant. Please supplement: Significance test and provide the $p$-value or confidence interval.

(6)	Please report the training time, the number of model parameters (total parameters), and the computational cost of each sample during inference (such as FLOPs or milliseconds), and compare them with the main baselines.

(7)	The provided code link in the manuscript doesn't work, resulting in a “The requested file is not found” error.

---

> ### Author Response · Authors · 2025-12-03
>
> We thank the reviewer for the constructive feedback and address the weaknesses and questions as follows.
>
> ---
>
> ### **(W1, Q1) Selection Bias in Brain Analysis**
> We appreciate the reviewer’s constructive comment regarding potential selection bias. While our initial intention was to isolate the most discriminative patterns by focusing on subjects the model classified correctly (thereby reducing noise from misclassified cases), we agree that this approach could inadvertently exaggerate group differences and limit generalizability.
>
> In response, we have revised the entire brain state analysis to include all test samples across all folds, ensuring an unbiased evaluation. Consequently, the results in **Section 5.3** and **Appendices F** have been updated to reflect this comprehensive analysis.
>
> Importantly, despite this broadening of the analysis scope, the core findings remain consistent. While we observed minor fluctuations in the overall values compared to the selective analysis, the dominant brain state patterns and transition dynamics specific to each diagnostic group remained prominent. To further validate the statistical reliability of these group differences, we performed independent two-sample $t$-tests on the transition probability maps(**Figure 3** and **Figure 11**). The analysis confirmed that the key transition patterns distinguishing the diagnostic groups are statistically significant ($p < 0.001$), verifying that our interpretability results are robust and generalizable across the entire test set.

---

> ### Author Response · Authors · 2025-12-03
>
> ### **(W2, Q2, Q3) Analysis and Justification of Orthonormal Clustering**
> We appreciate the reviewer’s detailed questions regarding the orthonormal clustering module. We have expanded our analysis to provide both theoretical justification and empirical evidence supporting this design choice.
>
> **1. Rationale and Analysis of Orthonormal Clustering (Addressing W2)**
> From a theoretical perspective, the advantage of employing orthonormal bases lies in their ability to maximize the discriminability of learned representations. As analytically derived in [1], orthonormal projections maximize the variance of cluster assignment probabilities. This constraint compels the cluster centers to be as distinct from each other as possible. Consequently, this prevents *mode collapse*, where multiple clusters might otherwise redundantly capture the same dominant signal, ensuring that the model learns non-redundant and informative latent states.
>
> To empirically validate this advantage and address the correspondence with brain functional modules, we visualized the relationship between our orthonormal clusters and the canonical Yeo-7 functional networks in **Section 5.4 (Figure 5)** of the revised manuscript. As shown in the figure, compared to randomly initialized learnable clusters, the assignment probability of our method reveals a distinct alignment where specific orthonormal clusters selectively map to distinct functional networks. In contrast, the random approach exhibits noisy, indiscriminative patterns or a strong bias toward a single cluster. This demonstrates that the orthogonality constraint effectively guides the model to disentangle complex brain dynamics into functionally coherent and non-redundant states, a property that is less pronounced in random or unconstrained clustering approaches.
>
> **2. Sensitivity to Initialization (Addressing Q2)**
> We employed the Gram-Schmidt process because it provides a constructive and numerically stable method to enforce strict orthogonality from random embeddings immediately at initialization. Unlike soft regularization constraints (e.g., adding an orthogonality penalty to the loss function), which only approximate orthogonality, Gram-Schmidt guarantees mathematically exact orthonormal bases without computational overhead.
>
> Regarding sensitivity to the initial vectors $\mathbf{V}$, we utilized Xavier uniform initialization to ensure a balanced variance distribution. Theoretically, while changing the initial $\mathbf{V}$ vectors rotates the absolute orientation of the basis vectors, the structural constraint of orthogonality remains invariant. Experimentally, our 5-fold cross-validation results demonstrate that the proposed method consistently achieves the best performance with statistical significance. Notably, the standard deviations for AUROC and ACC are generally lower than those of the second-best methods, suggesting that our model yields consistent performance across different data folds. However, acknowledging the reviewer's point regarding the randomness of initialization, we recognize the value of testing across diverse random seeds and plan to incorporate such robustness verification in future work.
>
> **3. Clarification on Fixed Orthonormal Cluster Centers (Addressing Q3)**
> In our "Orth (Ours)" mode, cluster centers are initialized once and **remain fixed** throughout training. This design choice guarantees strict orthogonality, which would otherwise be compromised if the centers were updated via gradient descent. This approach preserves the theoretical benefits discussed above, ensuring both **stability and interpretability**. By serving as stable *anchors*, these fixed orthonormal bases compel the model to map dynamic ROI embeddings ($\mathbf{h}_t$) into a structured, well-separated latent space. This encourages the learned representations to become discriminative and interpretable, effectively preventing the mode collapse often observed when cluster centers are allowed to drift during training.
>
> ### Reference
> > [1] Kan et al., “Brain network Transformer,” NeurIPS, 2022.

---

> ### Author Response · Authors · 2025-12-03
>
> ### **(W3, Q4) Numerical Stability and Constraints on SSM Parameterization**
> We appreciate the reviewer’s question regarding the numerical stability of the SSM parameterization. We clarify that our model builds upon the robust diagonal parameterization of structured SSMs (S4 [1], Mamba [2]), which inherently addresses these stability concerns through structural constraints and initialization.
>
> Regarding the stability of the inverse term $\mathbf{A}_r^{-1}$ (Eq. 10), the transition matrix $\mathbf{A}_r$ is structured as a **diagonal matrix** initialized with **strictly negative real eigenvalues** based on the HiPPO theory [1,3]. Since these eigenvalues are strictly negative and bounded away from zero, $\mathbf{A}_r$ is guaranteed to be non-singular, ensuring that the inverse operation is well-conditioned and does not induce numerical instability such as gradient explosion.
>
> With respect to the timescale parameter $\Delta_{r,t}$ and system stability (Eq. 11), we enforce strict positivity using a nonlinearity: $\Delta_{r,t} = \mathrm{softplus}(\epsilon_r + f_{\Delta}(x_{r,t}))$, with $\epsilon_r$ initialized within a stable range (i.e., $\mathrm{Uniform}([0.001, 0.1])$) following previous works [1,2]. Consequently, the interplay between the strictly negative eigenvalues of $\mathbf{A}\_r$ and the positive sampling step $\Delta_{r,t}$ ensures that the discrete transition matrix $\bar{\mathbf{A}}\_{r,t} = \exp(\Delta\_{r,t} \mathbf{A}\_r)$ always possesses eigenvalues within the unit interval $(0, 1)$. This guarantees that the system dynamics are contractive, leading to state decay rather than explosion, thereby ensuring asymptotic stability.
>
> For clarity and reproducibility, we have explicitly added these implementation details to **Appendix E.1** of the revised manuscript.
>
> ### References
> > [1] Gu et al., “On the parameterization and initialization of diagonal state space models,” NeurIPS, 2022.
> [2] Gu et al., “Mamba: Linear-time sequence modeling with selective state spaces," COLM, 2024.
> [3] Gu et al., “HiPPO: Recurrent memory with optimal polynomial projections,” NeurIPS, 2020.

---

> ### Author Response · Authors · 2025-12-03
>
> ### **(Q5) Statistical Relevance**
> We thank the reviewer for pointing out the need for statistical validation. To assess the statistical significance of the performance improvements, we conducted paired $t$-tests comparing our proposed method against the baseline models.
>
> We have updated the comparative tables (including Table 1) to explicitly indicate statistical significance. Specifically, we marked the results with an asterisk (*) where the performance improvement is statistically significant ($p < 0.05$). As shown in the revised tables, our method demonstrates significant improvements over most baselines, confirming the reliability of the performance gains.
>
>
>
> ### **(Q6) Computational Comparison**
> We appreciate the reviewer’s suggestion. Following the request, we have conducted a comprehensive computational analysis and updated **Appendix G.3** with the detailed results.
>
> As shown in the table, our model demonstrates parameter efficiency, requiring only 0.72M parameters (on ABIDE-I), which is significantly fewer than recent structured SSM baselines like FST-Mamba (33.69M).
> Regarding throughput, we observed that our model operates at a lower speed compared to some baselines. We attribute this to the computational overhead of the iterative global aggregation mechanism, which integrates spatial context into temporal updates at every step. However, we believe this trade-off is justified, as our primary contribution lies in the model's unique ability to explicitly capture dynamic brain state trajectories and provide neurobiological interpretability. We acknowledge the potential for optimization and plan to explore more efficient implementations of the aggregation module in future work.

---

> ### Author Response · Authors · 2025-12-03
>
> ### **(W4) Additional Dataset**
> We appreciate the reviewer’s suggestion to verify the model's generalization ability across diverse datasets.
>
> While our primary focus was on neurodevelopmental disorders (ASD and ADHD), we agree that testing on other conditions strengthens the evaluation. In response, we conducted additional experiments on the **COBRE** dataset [1] for schizophrenia diagnosis and reported the results in **Appendix G.1**.
>
> As shown in the new results, our model consistently achieved the best performance in AUROC, ACC, and SEN compared to the baselines. Notably, the improvement in ACC was statistically significant ($p < 0.05$) against most competing methods, demonstrating the effectiveness of our framework on the schizophrenia dataset. While these results substantiate the generalization capability of our framework, we acknowledge the further value of the datasets mentioned (e.g., ADNI, OASIS) and plan to extend our analysis to such a wider spectrum of disorders in future work.
>
> ### Reference
> > [1] Mayer et al., "Functional imaging of the hemodynamic sensory gating response in schizophrenia," Hum Brain Mapp., 2012.
>
>
> ### **(W5) Additional Competing Method**
> We appreciate the reviewer’s suggestion to broaden the comparative analysis.
>
> While our initial baseline selection prioritized recent Mamba-based architectures tailored for fMRI (e.g., fMRI-S4, FST-Mamba, BrainMAP), we agree that comparing against diverse architectures strengthens the evaluation. Accordingly, we have incorporated **ContrastPool** [1], a graph-based fMRI analysis method from the suggested list, into our experiments.
>
> We have included the implementation details and hyperparameter settings in **Appendix E.2**. The performance results for ContrastPool have been added to the main comparison table (**Table 1**) and the additional COBRE dataset analysis (**Table 4 in Appendix G.1**). As observed in the updated results, our model consistently yields superior performance compared to ContrastPool across most metrics. We will consider the other suggested architectures and plan to include a wider range of comparative methods in future work.
>
> ### Reference
> >[1] Xu et al., "Contrastive graph pooling for explainable classification of brain networks," TMI, 2024.
>
>
> ### **(W6) Additional Atlas**
> We appreciate the reviewer’s suggestion to assess the impact of different network scales.
>
> Following the comment, we extended our experiments on the ABIDE-I and ADHD-200 datasets using alternative parcellation schemes, specifically the **AAL atlas** [1] (116 ROIs) and the **Schaefer atlas** [2] (100 ROIs). We applied the same hyperparameter tuning procedure for these configurations, and the detailed results have been added to **Table 7 in Appendix G.2**.
>
> The results indicate that while our model maintains robust performance across different atlases, the configuration using the Yeo 2011 atlas yielded the best performance in our empirical evaluation. We acknowledge the importance of investigating a broader range of network scales and plan to include more comprehensive comparative analyses involving diverse atlases in future work.
>
>
> ### References
> > [1] Tzourio-Mazoyer et al., "Automated anatomical labeling of activations in SPM using a macroscopic anatomical parcellation of the MNI MRI single-subject brain," NeuroImage, 2002.
> [2] Schaefer et al., “Local-global parcellation of the human cerebral cortex from intrinsic functional connectivity MRI,”
> Cereb Cortex., 2018.

---

> ### Author Response · Authors · 2025-12-03
>
> ### **(Q7) Link Error**
> We apologize for the oversight regarding the broken link. We have updated the URL in the revised manuscript and verified that the anonymous repository is now fully accessible.

---

### Official Review · Reviewer_AGHL · 2025-11-02

**Soundness:** 3
**Presentation:** 3
**Contribution:** 2
**Rating:** 4
**Confidence:** 3

**Summary:**

The paper introduces DyBraSS, a state-space model designed for analyzing dynamic brain states from resting-state fMRI data. The core idea is to create a model that handles both spatial and temporal aspects of brain dynamics simultaneously. It uses a "global aggregation module" with an orthonormal clustering approach to group evolving brain activity into interpretable states. The method is evaluated on the ABIDE-I and ADHD-200 datasets for diagnostic classification, where it is shown to perform better than several existing methods. The paper also provides an analysis of the learned brain states, suggesting they align with known neurobiological patterns in ASD and ADHD.

**Strengths:**

1. The proposed method for jointly modeling spatial and temporal dynamics look good to me. Directly integrating a global context into the ROI-level updates, rather than treating them as an ordered sequence, is a more faithful representation of brain topology.
2. The experimental evaluation is quite thorough. The model is compared against a good range of recent SOTA baselines from different architectural families (CNN, Transformer, SSM) on two standard, multi-site datasets. The performance gains shown in Table 1 are clear.
3. The attempt to make the model's outputs interpretable is a significant plus. The brain state analysis in Section 5.3, particularly the visualization of state transitions and network configurations (Figures 3 and 4), connects the model's learned patterns back to clinical neuroscience, which is often missing in purely performance-driven machine learning papers.

**Weaknesses:**

1. The motivation for using orthonormal clustering could be stronger. While prior work is cited (lines 280-282), the paper doesn't fully explain why orthogonality is a necessary or superior constraint for defining brain states compared to other clustering approaches. The ablation study (Table 2) shows it works best among the tested options, but the underlying reason isn't entirely clear.
2. The model's complexity seems quite high. With stacked DynBrain-Mamba blocks, multiple MLPs, and several moving parts (lines 769-774), it's hard to tell which components are doing the heavy lifting. The parameterization also involves many choices (e.g., number of clusters, state dimensions) that might be sensitive and difficult to tune.
3. The clinical interpretations, while interesting, feel a bit speculative. For instance, attributing the State 6→4 transition in ASD to "altered coordination between sensorimotor processing and cognitive control systems" (lines 455-456) is a strong claim based on correlational data. While the interpretation aligns with existing literature, the link could be drawn more cautiously.

**Questions:**

1. Regarding the global aggregation module (Section 4.2.2): The use of orthonormal bases for cluster centers is an interesting choice. Could the authors elaborate on the neurobiological intuition behind this constraint? Does forcing the cluster centers to be orthogonal impose a structure that is believed to exist in brain functional organization, or is this primarily a choice that was found to be effective empirically?
2. In the ablation study (Table 2), the "Attention" aggregation method performs worse than the clustering-based methods. This is somewhat surprising given the success of attention mechanisms in related fields. Could the authors provide some insight into why this might be the case? Was it a simple attention mechanism, and is it possible a more sophisticated variant could have been more competitive?
3. The paper mentions that the learned brain states align with known neurobiological alterations (lines 089-090). In the ABIDE-I analysis, the TC group shows a tendency to remain in State 1, described as reflecting "stable engagement of internally oriented cognition" (lines 465-466). Could the authors clarify what the functional consequence of not remaining in this state might be for the ASD group? Is the model suggesting that the ASD group is less able to sustain this mode of brain activity?
4. For the dFC calculation (lines 200-204), a sliding-window approach is used. The choice of window length and stride can significantly impact the resulting dynamics. While the paper normalizes these based on TR, how sensitive is the model's performance to the target window size (15s) and stride (3s)? Was any analysis done to select these specific values?

---

> ### Author Response · Authors · 2025-12-03
>
> We thank the reviewer for the constructive feedback and address the weaknesses and questions as follows.
>
> ---
> ### **(W1, Q1) Justification and Neurobiological Intuition of Orthonormal Clustering**
>
> We appreciate the reviewer for raising this fundamental question regarding the motivation behind our clustering approach. We clarify that while orthogonality does not imply that the brain's functional organization is strictly orthogonal, it serves as an effective structural constraint that reflects key characteristics of functional brain networks. We address this based on both theoretical justification and empirical evidence.
>
> **1. Theoretical & Neurobiological Motivation**
> The choice of orthonormal bases is motivated by both statistical learning properties and the modular nature of brain functional organization.
>
> Theoretically, as analytically shown in [1], orthonormal projections maximize the variance of cluster assignment probabilities. This constraint forces the learned latent states (cluster centers) to be as distinct from each other as possible, preventing *mode collapse* where multiple clusters might otherwise redundantly represent the same dominant signal.
>
> Neurobiologically, while functional brain networks exhibit dynamic interactions and transient overlaps, they are fundamentally characterized by intra-network coherence and inter-network distinctiveness [2,3]. By enforcing orthogonality on the cluster centers, we impose a structural constraint that encourages the model to learn these distinct functional modules. Importantly, this does not rigidly force brain states to be orthogonal; rather, the orthogonal centers serve as distinct *anchors,* while our soft assignment mechanism allows the model to flexibly capture the continuous and overlapping transitions inherent in brain dynamics. This ensures that the resulting states are discriminative and interpretable, avoiding ambiguous or redundant representations.
>
>
> **2. Empirical Verification: Correspondence with Functional Networks**
> To empirically validate whether this constraint captures biological reality, we analyzed the correspondence between our learned orthonormal clusters and the canonical Yeo-7 functional networks. These results have been added to **Section 5.4** of the revised manuscript.
>
> As shown in **Figure 5**, we visualized the assignment probabilities with respect to the cluster centroids. We observed a clear alignment where specific orthonormal clusters correspond to distinct functional networks. This pattern is significantly more pronounced compared to the random initialization strategy, which resulted in noisy representations with weak network specificity. This empirical comparison suggests that the orthonormal constraint encourages the model to capture functionally coherent patterns, potentially reflecting the intrinsic modularity of brain organization.
>
> ### References
> > [1] Kan et al., “Brain network Transformer,” NeurIPS, 2022.
> [2] Fan et al., "Altered connectivity within and between the default mode, central executive, and salience networks in obsessive-compulsive disorder," J. Affect. Disord., 2017.
> [3] Xi et al., "Triple network hypothesis-related disrupted connections in schizophrenia: A spectral dynamic causal modeling analysis with functional magnetic resonance imaging," Schizophr Res. 2021.

---

> > ### Author Response · Authors · 2025-12-03
> >
> > ### **(W2) Model Complexity**
> > We appreciate the reviewer’s feedback on the model architecture. We clarify that each component is designed to address specific spatiotemporal challenges of fMRI data, and empirical evidence confirms that the model remains computationally efficient, particularly regarding model size and parameter efficiency.
> >
> > **1. Contribution of Components (Heavy Lifting)**
> > As demonstrated in our ablation study (**Table 2**), removing the global aggregation module resulted in a significant performance drop. This confirms that the interaction between local dynamics and the global clustering mechanism is not redundant but plays a critical role in integrating spatial context into temporal evolution.
> >
> > **2. Parameter Efficiency and Hyperparameter Tuning**
> > To address the concern about complexity, we conducted a computational cost analysis comparing our model with baselines. The results, added to **Appendix G.3**, show that despite the stacked architecture, our model remains highly compact. Specifically, on the ABIDE-I dataset, our model requires only **0.72M parameters**, which is significantly more efficient than recent structured SSM baselines like FST-Mamba (33.69M) and comparable to lightweight CNN-based methods. This indicates that the design choices contribute to performance without causing parameter bloat.
> >
> > Regarding hyperparameter tuning, we primarily searched the number of clusters $K \in \\{8, 16, 32\\}$, the MLP projection dimension $D \in \\{128, 256, 512\\}$, and the classifier hidden dimension $H \in \\{4, 8\\}$. As detailed in **Appendix E.2**, this search space is comparable to or smaller than that of the baseline methods, indicating that our model does not require excessive tuning compared to existing approaches.

---

> ### Author Response · Authors · 2025-12-03
>
> ### **(Q2) Ablation study about "Attention" Aggregation**
> We appreciate the reviewer’s insight regarding the performance of the attention mechanism. As the reviewer noted, while attention is highly effective in many domains, our empirical results indicated that it was less suitable for modeling dynamic brain states in this specific context compared to clustering-based approaches.
>
> In our ablation study, we evaluated two common attention-based aggregation strategies: (1) using a learnable classification token as the global embedding, and (2) using the attention map directly as the global embedding. We reported the results from the latter as it showed better performance.
>
> We interpret the superior performance of our proposed clustering method as stemming from its ability to structure ROI dynamics into distinct latent bases. While the attention map captures dense pairwise relationships between all regions, our clustering approach organizes ROI dynamics based on a compact set of orthonormal bases. This structural constraint allows the model to extract clearer and more discriminative global embeddings, which appears to be advantageous for classifying temporal dynamics in fMRI data.

---

> ### Author Response · Authors · 2025-12-03
>
> ### **(W3, Q3) Neurobiological Interpretation**
> We appreciate the reviewer’s cautious perspective regarding the neurobiological interpretation of our results.
>
> **1. Refinement of Clinical Interpretations (Addressing W3)**
> We agree with the reviewer that the interpretation regarding cognitive deficits requires caution. Accordingly, while noting that our interpretation was grounded in established literature describing sensory-executive imbalances in ASD, we have softened the language in the revised manuscript to present these findings as patterns "consistent with" or "potentially reflecting" altered coordination, rather than definitive evidence of underlying neural mechanisms.
>
> **2. Functional Implications of State 1 Stability in ASD (Addressing Q3)**
> We confirm the reviewer's intuition regarding State 1, which characterizes stable internally oriented cognition through the co-activation of default mode, somatomotor, and visual networks. Quantitatively, our fractional dwell time analysis (**Figure 9**; **Appendix F.3.1**) demonstrates that the TC group exhibits significantly longer persistence in this state compared to the ASD group ($p < 0.05$). This finding aligns with the "default mode network instability" hypothesis reported in autism literature [1,2], suggesting that the reduced stability of State 1 in ASD reflects a disruption in intrinsic neural processing, potentially contributing to deficits in cognitive flexibility and sensory modulation.
>
>
> ### References
> >[1] He et al., "Dynamic functional connectivity analysis reveals decreased variability of the default-mode network in developing autistic brain," Autism Res., 2018.
> [2] Padmanabhan et al., "The default mode network in autism," Biol Psychiatry Cogn Neurosci Neuroimaging, 2017.

---

> ### Author Response · Authors · 2025-12-03
>
> ### **(Q4) Variants for dFC representation**
> We thank the reviewer for raising this point regarding the robustness of our dFC parameter selection. We conducted a ablation studies by varying the target window length ($w_{\text{sec}}$) and stride ($s_{\text{sec}}$). The detailed results have been added to **Appendix G.2** of the revised manuscript.
>
> As shown in the experiments, we observed a general trend where performance deteriorates as the window length and stride increase. We attribute this to the fact that larger windows and strides likely over-smooth the temporal dynamics, thereby failing to capture the rapid, spontaneous transitions in brain states that are critical for diagnosis. Consequently, the configuration of $w_{\text{sec}}=15$ and $s_{\text{sec}}=3$ yielded the best performance in our empirical evaluation, which served as the basis for the hyperparameters used in our main report.

---

### Author Response · Authors · 2025-12-03
**Summary of Revisions**

We sincerely appreciate the reviewers’ valuable comments and constructive suggestions, which were helpful in improving the quality of our manuscript.

In response to these suggestions, we have significantly revised the manuscript. The major updates are summarized below:

**1. In-depth Analysis of Clustering Assignment**
To empirically validate the effectiveness of the proposed orthonormal clustering, we added a comparative visualization in Section 5.4 (Figure 5). The results demonstrate that our orthonormal approach captures distinct, functionally coherent brain states, whereas learnable random initialization leads to ambiguous or redundant patterns.

**2. Theoretical Stability Analysis**
To address the stability of the proposed SSM formulation, specifically regarding the added global aggregation term, we provided a formal analysis in Appendix B. We demonstrate that the global aggregation is Lipschitz-continuous with respect to the hidden states and acts as a controlled perturbation of the asymptotically stable local dynamics. This suggests that the resulting closed-loop system is resistant to unbounded growth of the states in practice.

**3. Robustness of Brain Analysis**
We revised our analysis protocol to include all test subjects across all folds (updated in Section 5.3 and Appendix F). Consequently, the core findings remained consistent with our initial results. We further validated the group differences in transition matrices using independent-samples t-tests ($p < 0.001$), confirming the statistical reliability of the identified neurobiological patterns.

**4. Expanded Experiments for Generalization**
To verify generalization capability, we extended our experiments to an independent psychiatric dataset, COBRE (schizophrenia), and included an additional graph-based baseline, ContrastPool. In addition, we supplemented Appendix G.2 with additional ablation results covering dFC representation variants and comparisons across different atlases. As a result, DyBraSS consistently achieved superior performance on the COBRE dataset and against the added baseline. We also verified robustness across different atlases (AAL, Schaefer) and dFC variants.

**5. Computational Efficiency Analysis**
We added a computational comparison of model parameters and inference throughput in Appendix G.3. Our model demonstrates high parameter efficiency (0.72M parameters), being significantly more compact than recent SSM baselines (e.g., FST-Mamba: 33.69M). Although the aggregation mechanism incurs a computational overhead, we consider this a worthwhile investment for enabling explicit trajectory modeling and interpretability. We plan to explore more efficient architectural variants in future work.

**6. Methodological Clarifications and Visual Enhancements**
We added a structured Algorithm Box (Pseudo-code) in Appendix C to clarify the data flow and module interactions. Additionally, we improved the brain state transition diagrams by incorporating probabilistic values for better interpretability.

We believe these additional experiments and clarifications strongly reinforce the validity and robustness of our proposed method, DyBraSS, which is based on a novel structured SSM formulation that explicitly models evolving brain state trajectories and enhances clinical interpretability.

---

### Meta-Review · Area_Chair_oXNU · 2026-01-09

**Summary:**

The four reviewers’ evaluations were largely negative. The main reasons were the paper’s insufficient originality and weak theoretical grounding, including unclear motivation for orthonormal clustering, lack of identifiability analysis and rigorous stability guarantees, as well as overreliance on informal claims. Reviewers also identified serious weaknesses in experimental design and generalization, such as selection bias in early analyses, limited additional datasets, and the absence of key benchmarks repeatedly requested. Furthermore, the paper was criticized for over-speculating on interpretability and clinical relevance without causal or neurobiological validation, and for shortcomings in clarity, efficiency reporting, and reproducibility. Together, these unresolved issues significantly undermined the credibility and impact of the work, resulting in a rejection decision.

**Reviewer Concerns:**

Resolved Issues:
- Addressing criticisms of dXQf and zr9w, the authors removed selection bias, now include all test samples and all folds, and added t-tests (p < 0.001 for group differences, p < 0.05 for performance improvements). This significantly improves the reliability of the results.

- For AGHL, dXQf, and fFeQ, the authors added visualizations in Section 5.4 and Figure 5, demonstrating the alignment of orthogonal clustering with the Yeo-7 network and explaining its superiority over attention mechanisms (structured embeddings vs. pairwise attention).

- For dXQf and fFeQ, the authors provided proofs of Lipschitz continuity and contraction mapping in Appendix B, ensuring the stability of the feedback loop; they also explained the positivity of Δ_r,t and the application of softplus.

- ContrastPool baseline was added, and AAL/Schaefer atlases were tested (Appendix G.1-2). Computational efficiency was also reported (0.72M parameters, superior to 33.69M for FST-Mamba, Appendix G.3).


Remaining Issues:
- Speculative Nature of Clinical Interpretation: Although concerns about AGHL were softened, brain state analyses (such as the instability of State 1 in ASD) are still based on correlational data and lack causal validation or more neurobiological citations. This may still require further empirical support.

- Model Complexity and Sensitivity: While the hyperparameter sensitivity and complexity mentioned by AGHL have been ablated, their impact on practical deployment (e.g., in resource-constrained environments) has not been fully quantified.

- Broader Generalization: Although COBRE has been added, the third dataset recommended by fFeQ (such as ADNI for Alzheimer) is not covered; cross-network scale analysis (dXQf) also only covers a portion of the addresses, and validation on more mental illness datasets may still be needed.

**Reviewer Scores:**

Reviewer AGHL (Initial 4): Their concerns (such as clustering motivation and interpretability) have been directly addressed by visualization and softened language; stability is not their primary concern, but the overall improvement strengthens their confidence. I think they will raise the score to 6-7 (marginally above) as the performance and interpretability advantages are reinforced, moving towards weak acceptance.

Reviewer dXQf (Initial 2): Bias, stability, generalization, and comparison have been fully addressed (all sample analyses, new datasets, proofs, t-tests). Given their high confidence (5), they may acknowledge the improvement if the discussion confirms consistency, but remain cautious about remaining generalization concerns. I expect the score to rise to 4-5 (marginally below), moving from strong rejection to borderline.

Reviewer zr9w (Initial 6): Their main concerns (bias and significance) have been completely addressed (full sample + t-tests). As an already positive reviewer, the discussion will further validate the link to neurobiology, and I think the score will rise to 7-8 (accept), reinforcing their strengths.

Reviewer fFeQ (Initial 2): This is the most critical, with 15+ points, but it addresses most issues (such as stability, pseudocode, efficiency, and dataset), though not all (such as all references). If the discussion allows for clarification of the remaining design choices, their rating might rise from a strong rejection to 4 (marginally below), acknowledging novel contributions but still concerned about clarity.

---

### Decision · Program_Chairs · 2026-01-26

Reject